# TAVRNN: Temporal Attention-enhanced Variational Graph RNN Captures Neuronal Dynamics and Behavior

## Abstract

We introduce Temporal Attention-enhanced Variational Graph Recurrent Neural Network (TAVRNN), a novel framework for analyzing the evolving dynamics of neuronal connectivity networks in response to external stimuli and behavioral feedback. TAVRNN captures temporal changes in network structure by modeling sequential snapshots of neuronal activity, enabling the identification of key connectivity patterns. Leveraging temporal attention mechanisms and variational graph techniques, TAVRNN uncovers how connectivity shifts align with behavior over time. We validate TAVRNN on two datasets: *in vivo* calcium imaging data from freely behaving rats and novel *in vitro* electrophysiological data from the *DishBrain* system, where biological neurons control a simulated environment during the game of *pong*. We show that TAVRNN outperforms previous baseline models in classification, clustering tasks and computational efficiency while accurately linking connectivity changes to performance variations. Crucially, TAVRNN reveals that high game performance in the *DishBrain* system correlates with the alignment of sensory and motor subregion channels, a relationship not evident in earlier models. This framework represents the first application of dynamic graph representation of electrophysiological (neuronal) data from *DishBrain* system, providing insights into the reorganization of neuronal networks during learning. TAVRNN's ability to differentiate between neuronal states associated with successful and unsuccessful learning outcomes, offers significant implications for real-time monitoring and manipulation of biological neuronal systems.

## 1 Introduction

The field of artificial intelligence has from the outset used natural systems, refined over evolutionary timescales, as templates for its models (1). Neuroscience has been a significant source of inspiration, from the McCulloch-Pitts neuron and the parallel distributed architectures of connectionism and deep learning, to the contemporary call for Neuro-AI as a paradigm for research in AI (2). Progress leveraging the neurocomputational capacity of biological neurons requires more advanced machine learning methods to enable better prediction and interpretation of behavior from neuronal activity. The understanding gained from these efforts may offer the potential for more refined machine learning algorithms that require less data and energy.

Past attempts to examine higher-order neuronal dynamics typically isolate the temporal evolution of neuronal signals (3; 4; 5). However, the specific network dynamics integral to the neuronal learning process, particularly the unit-population relationship, have yet to be fully explored. Analysis at either level can be informative but fail to explain behavioral outcomes sufficiently (6; 7). To address this gap we analyzed the spiking activity at the single unit level of *in vivo* calcium imaging data from the hippocampus of freely behaving rats (8) and *in vitro* electrophysiological data from the *DishBrain* system (6). Within the *DishBrain* framework, *in vitro* neuronal networks are intricately combined with *in silico* computing via high-density multi-electrode arrays (HD-MEAs). Through real-time closed-loop structured stimulation and recording, these biological neuronal networks (BNNs) are then embedded in a simplified Pong-game and showcase self-organized adaptive electrophysiological dynamics. We propose a novel approach: investigating the temporal trajectories of a single neuron data in synchronization with the online evolution of behavior. Exploring the evolving structure and functional connectivity of BNN in this integrated manner, we aim to provide a more comprehensive understanding of the neuronal mechanisms driving adaptive learning in real-time environments.

By analyzing the simultaneous evolution of neuronal and behavioral data, this method reveals crucial insights into the links between population-level neuronal activity and behavior. Moreover, it extends beyond this scope by examining interactions between individual neurons and uncovering the patterns

that underlie learning and neuronal information processing in a system such as the *DishBrain* system. The dynamic interplay between neurons within the network not only facilitates information processing and response generation but also reveals how learning modulates synaptic interactions, affecting signal transmission across the network. This approach enhances our understanding of both cellular and network-level processes critical to learning, with significant implications for neuroscience and artificial intelligence. It also holds promise for informing the development of advanced learning algorithms and innovative treatments for neurological disorders.

## 2 BACKGROUND

### 2.1 LARGE-SCALE NEURONAL RECORDINGS AND LEARNABLE LATENT EMBEDDINGS TO LINK BRAIN AND BEHAVIOR

Simultaneous recordings from large neuron populations offer rich electrophysiological data crucial for understanding brain function. A key challenge in neuroscience is linking these high-dimensional recordings to neurocomputational processes and ensuing behavior, a task that spans a wide range of recording schemes and datasets. In this work, we utilize two exemplar datasets: a high-density microelectrode arrays (HD-MEA) recordings of *in vitro* neurons and hippocampal data from behaving rats, allowing us to explore the connection between neuronal dynamics and behavior across different scales (9). Progress in Synthetic Biological Intelligence (SBI) requires innovative methods to analyze neuronal data and link brain function to behavior. Network models allow the study of simultaneous recordings from biological neuronal networks (BNNs), emphasizing the role of neuronal assemblies in memory (10) and stimulus processing (11). Although neuronal latent embeddings offer insights into behavior-related neuronal correlates, there is a paucity of nonlinear techniques that can adeptly and flexibly utilize combined behavioral and observed neuronal data to elucidate the underlying neuronal dynamics. Conversely, existing nonlinear methods for associating neuronal and behavioral data, in a single model, usually investigate the temporal trajectory of the entire neuronal population as a whole, neglecting the interaction-based network of single neurons. These methods also struggle to track individual neuron activity and uncover the evolving connectivity that facilitates adaptive learning (3). Population-wide analysis of neuronal recordings demands a novel theoretical framework for advancing the algorithmic understanding of intelligence.

### 2.2 NODE EMBEDDING TECHNIQUES

Node embedding techniques translate network nodes into vectors within a low-dimensional latent space, enabling traditional vector-based machine learning methods (12). Current approaches typically treat networks as static, assuming fixed node and edge sets throughout the learning process (13; 14; 15; 16; 17; 18). These methods often apply static embeddings to network snapshots, which simplifies the inherently time-varying nature of neuronal dynamics and the resulting temporal network dependencies, potentially overlooking the evolving characteristics of neuronal networks (19). Several techniques have been developed to account for the temporal evolution of networks (20; 21; 22; 23), but they often represent each node with a deterministic vector in a low-dimensional space (24), failing to capture the uncertainty in node embeddings that arises from integrating node attributes and network structure. This limitation underscores the need for probabilistic embedding techniques that reflect the uncertain, dynamic nature of node characteristics and interactions over time.

To address these shortcomings, the Graph Recurrent Neural Network (GRNN) was proposed to extend traditional graph convolutional networks to dynamic networks (25). However, GRNN struggled to fully capture the complex interaction between network topology and node attributes due to its reliance on unimodal distributions. To improve the modeling of sparse dynamic networks, the Variational Graph Recurrent Neural Network (VGRNN) (26) was introduced, but it still faced challenges in emphasizing relevant historical information and distinguishing the varying importance of past time steps. Our model enhances GRNN by incorporating high-level latent random variables, providing richer and more interpretable latent representations. We propose an improvement to the VGRNN framework by introducing a temporal attention mechanism that evaluates the topological similarity of the network across time steps, accounting for varying time lags to better capture complex network dynamics. This approach provides a deeper understanding of how network structures evolve over time and, in systems like *DishBrain* , offers insights into the neuronal mechanisms driving adaptive learning in *in vitro* neuronal assemblies.

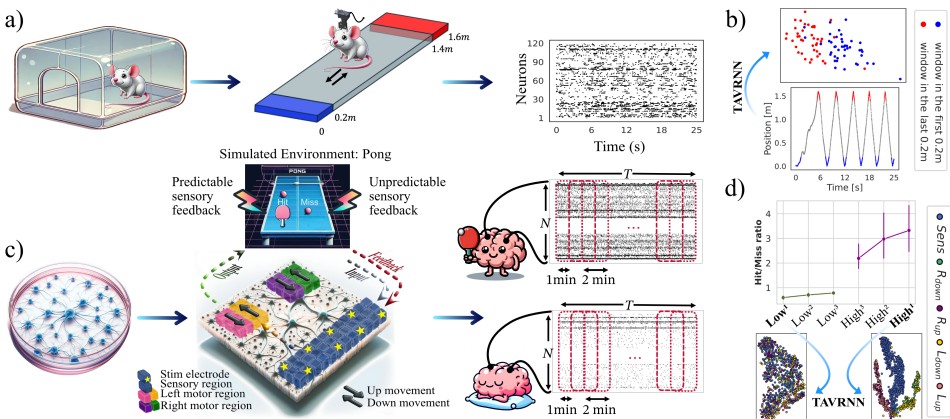

Figure 1: **a)** Schematic of rat hippocampus data collection and neuronal data over a 25-second window during traversal of a 1.6 m track. The track's ends are color-labeled in the behavioral plot, showing the rat's position. **b)** The low-dimensional neuronal data representation for the two ends using TAVRNN. **c)** Schematic illustration of the *DishBrain* feedback loop, game environment, and electrode configurations. Sample Gameplay and Rest session spike rasterplots are shown from $N = 900$ electrodes. **d)** Hit/miss ratio for the three top and bottom performing windows during gameplay averaged over all cultures and lower-dimensional representation of the neuronal data for a sample culture for the best ($High^1$) and worst ($Low^1$) performing windows using TAVRNN.

## 3 DATASETS

**Rat hippocampus dataset**    We used the dataset from (8), consisting of multicellular recordings from 120 putative pyramidal neurons in the CA1 hippocampal subfield of male Long–Evans rats using silicon probes. Rats ran on a 1.6-meter linear track, receiving water rewards at both ends (Fig. 1a), with spiking data recorded at 40 Hz for 254 seconds. The rat's position on the track was simultaneously recorded (Fig. 1b) and served as ground truth to validate TAVRNN in a downstream classification task to link population neuronal activity to the rat's position on the track, which, based on previous evidence, is thought to be encoded by place cells in the hippocampus (27).

*DishBrain* **cell culture dataset**    The *DishBrain* system, integrated in real-time with the MaxOne MEA software (Maxwell Biosystems, AG, Switzerland), facilitates closed-loop stimulation and recording of cultured cortical networks during engagement in a simplified version of Pong (6). Neuronal activity from 24 cultures across 437 sessions (262 'Gameplay', 175 'Rest') was recorded at 20 kHz using an HD-MEA with 900 channels. During Gameplay, sensory stimulation was delivered via 8 electrodes using rate coding (4Hz–40Hz) for the ball's $x$-axis and place coding for the $y$-axis. Paddle movement was controlled by the level of electrophysiological activity in counterbalanced "motor areas" (Fig. 1c). In the "motor regions," activity in half of each subregion moved the paddle "up" ($L_{up}$, $R_{up}$) and the other half moved it "down" ($L_{down}$, $R_{down}$). Cultures received feedback via the same sensory regions, such that unpredictable 150 mV stimulations at 5 Hz were introduced when they missed the ball as random external inputs into the system. This was applied to arbitrary locations among the 8 sensory electrodes, at varied intervals lasting up to 4 seconds. A configurable 4-second rest period ensued before the next rally commenced. During Rest sessions, activity was recorded to move the paddle without stimulation or feedback, while outcomes were still recorded. Gameplay and Rest sessions lasted 20 and 10 minutes, respectively, with spiking events from all channels extracted in each session. Further details on this system are provided in Appendix A.1, A.2, A.3. Behavioral data was collected by measuring the cultures' ability to intercept the ball, quantified by the number of 'hits'. Each rally ended with a 'miss', resetting the ball to a random position for a new episode. The hit/miss ratio was defined as the ratio of accurate hits to the number of missed balls (i.e. number of rallies played). This dataset was used in a downstream clustering task with regions applied as labels to observe how channels clustered at different performance levels.

**Preprocessing**    For the rat hippocampal recording, we used binary spiking data from 120 neurons across 10,178 time points at 40 Hz. We selected time windows of spiking activity when the rat was within the first and last 0.2 meters of the track, yielding 85 crossings (Fig. 1b). These varying length time windows were subsequently labeled as 1 for the beginning and -1 for the end of the track for the downstream classification task. To ensure that the covariance matrix is not ill-conditioned in

these time windows, according to the Marchenko-Pastur distribution (28; 29; 30), we compared it to a shuffled control, preserving neuron identity while shuffling time points independently. This process was repeated 1000 times to estimate confidence intervals, considering only correlations beyond the 95% confidence bounds in the analysis. For further details, see Appendix A.6.

For each of the 24 neuronal cultures in the *DishBrain* system, spiking activity from all Gameplay and Rest trials was down-sampled from a sampling frequency of 20KHz by applying a binary OR operation within 50 ms time bins. A value of 1 was assigned if a spike occurred in any trial within the bin, and 0 otherwise. This process produced 24 binary spiking time series (one per culture), each with 900 channels, and 24,000 time points during Gameplay and 12,000 during Rest. To investigate the single-unit interactions and dynamics of the underlying neuronal networks and their variations in game performance, we then segmented each Gameplay or Rest session into sliding windows of 2 minutes, each overlapping by half a window (i.e., 1 minute). This method generated 19 snapshots during Gameplay and 9 during Rest sessions. The selected window size ensured that the covariance matrices were not ill-conditioned based on Marchenko-Pastur distribution from random matrix theory (28). We computed the hit/miss ratio for each time window by averaging results across all trials for each culture. The three time windows with the highest and lowest hit/miss ratios were classified respectively as the best ($High^{1,2,3}$) and worst ($Low^{1,2,3}$) performing windows. $High^{1,2,3}$ were chosen for the main comparative analyses in the following sections (see Fig. 1d for average performance levels in these six time windows and Appendix A.4 for additional comparisons).

# 4 METHODOLOGY

## 4.1 TEMPORAL NETWORK CONSTRUCTION

Within each window of either dataset, we constructed a network adjacency matrix representing functional connectivity using zero-lag Pearson correlations as edges and 120 neurons or 900 channels as nodes. We employed graph kernels for selecting the connectivity inference method (Pearson correlation) and determining the cutoff threshold for the *DishBrain* dataset (see Appendix A.5).

The functional connectivity between nodes from both datasets was represented as edges in a matrix. For each time window $t$, the corresponding temporal network is represented by a graph $G_t \equiv (V, E)$, where $v_i \in V$ represents a specific channel, and $e_{ij} \in E$ denotes the connectivity edge between nodes $v_i$ and $v_j$. The structure of these dynamic network graphs $G_t$ is captured in time-resolved adjacency matrices $\mathbf{A}_t = [a_{t,ij}]$, with elements in $\{0,1\}^{N \times N}$, where $N$ is the number of nodes. These matrices are generated by applying a threshold (as obtained from the graph kernels - see Appendix A.5) to the functional connectivity matrices, retaining only the connections above that threshold based on absolute correlation values and setting the remainder to zero. Note that given this input structure, TAVRNN is capable of handling temporal graphs from time windows of varying lengths as in the rat hippocampal dataset in this study. Additionally, each dynamic graph $G_t$ includes node features $\mathbf{X}_t = [x_{t,1}, \ldots, x_{t,N}]^\top$ in $\mathbb{R}^{N \times D}$, where $x_{t,i}$ corresponds to the feature vector of each node $v_i$, calculated from the connection weights of each node and $D$ is the number of features.

## 4.2 TEMPORAL ATTENTION-ENHANCED VARIATIONAL GRAPH RNN (TAVRNN)

In this section, a probabilistic TAVRNN framework is developed to extract representative latent embeddings of the dynamic connectivity networks in a purely unsupervised manner. Fig. 2 summarises the pipeline of the introduced framework in this section. The Python implementation of our proposed framework is available at the following Github Repository.

### 4.2.1 SPATIOTEMPORAL VARIATIONAL BAYES

We present a spatiotemporal variational Bayes objective function designed to maximize the lower bound on the log model-evidence known as the evidence lower bound (ELBO) written as $\log p_\theta(\mathcal{A}|\mathcal{X})$ or equivalently minimize its negative value known as the variational free energy (VFE). This objective is applied to a series of adjacency matrices $\mathcal{A} = \{\mathbf{A}_t\}_{t=0}^T$ from dynamic networks, based on the sequence of node features $\mathcal{X} = \{\mathbf{X}_t\}_{t=0}^T$, where $T$ is the length of the sequence. Introducing a latent embeddings sequence $\mathcal{Z} = \{\mathbf{Z}_t\}_{t=0}^T$, the VFE $\mathcal{L}^{VFE}(\theta, \phi)$ can be written via importance decomposition as:

$$\mathcal{L}_{VFE}(\theta, \phi) = -\mathbb{E}_{q_\phi(\mathcal{Z}|\mathcal{X}, \mathcal{A})} \left[ \log \frac{p_\theta(\mathcal{A}, \mathcal{Z}|\mathcal{X})}{q_\phi(\mathcal{Z}|\mathcal{X}, \mathcal{A})} \right]. \tag{1}$$

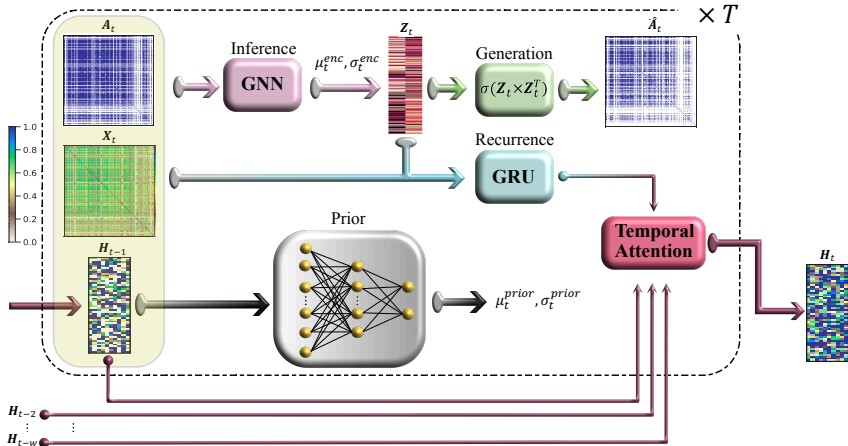

Figure 2: A schematic illustration of the TAVRNN framework.

Here, the subscripts $\theta$ and $\phi$ represent the parameters of the GNN that model the generative distribution $p_\theta(\mathcal{A}, \mathcal{Z}|\mathcal{X})$ and the posterior distribution $q_\phi(\mathcal{Z}|\mathcal{X}, \mathcal{A})$, respectively. Using the following general ancestral factorizations:

$$p_\theta(\mathcal{A}, \mathcal{Z}|\mathcal{X}) = \prod_{t=0}^{T} p_\theta(\mathbf{A}_t|\mathbf{Z}_{\leq t}, \mathcal{X}, \mathbf{A}_{<t}) \times p_\theta(\mathbf{Z}_t|\mathcal{X}, \mathbf{A}_{<t}, \mathbf{Z}_{<t}), \tag{2}$$

$$q_\phi(\mathcal{Z}|\mathcal{X}, \mathcal{A}) = \prod_{t=0}^{T} q_\phi(\mathbf{Z}_t|\mathcal{X}, \mathbf{A}_{\leq t}, \mathbf{Z}_{<t}), \tag{3}$$

Eq. equation 1 is expanded to yield the sequential VFE (sVFE) as follows:

$$
\begin{aligned}
\mathcal{L}^{sVFE}(\theta, \phi) = & -\sum_{t=0}^{T} \Big[ \mathbb{E}_{q_\phi(Z_{\leq t}|X, A_{\leq t}, Z_{<t})} \big[ \log p_\theta(\mathbf{A}_t|\mathbf{Z}_{\leq t}, \mathcal{X}, \mathbf{A}_{<t}) \big] \\
& + \mathcal{D}^{KL} \big[ q_\phi(\mathbf{Z}_t|\mathcal{X}, \mathbf{A}_{\leq t}, \mathbf{Z}_{<t}) \| p_\theta(\mathbf{Z}_t|\mathcal{X}, \mathbf{A}_{<t}, \mathbf{Z}_{<t}) \big] \Big].
\end{aligned}
\tag{4}
$$

Here, $\mathbf{A}_{\leq t}$ and $\mathbf{A}_{<t}$ refer to the partial sequences up to the $t^{th}$ and $(t-1)^{th}$ time samples, respectively. $\mathcal{D}^{KL}$ represents the (positive-valued) Kullback-Leibler divergence (KLD).

Since we want $\mathbf{Z}_t$ to represent all the information of $\mathbf{A}_t$, we replace $p_\theta(\mathbf{A}_t|\mathbf{Z}_{\leq t}, \mathcal{X}, \mathbf{A}_{<t})$ in Eq. equation 4 by $p_\theta(\mathbf{A}_t|\mathbf{Z}_t)$. Noting that Eq. equation 4 holds for any arbitrary density function $q_\phi$, we restrict our options to the density functions that satisfy the following equation:

$$q_\phi(\mathbf{Z}_t|\mathcal{X}, \mathbf{A}_{\leq t}, \mathbf{Z}_{<t}) = q_\phi(\mathbf{Z}_t|\mathbf{X}_{\leq t}, \mathbf{A}_{\leq t}, \mathbf{Z}_{<t}) \tag{5}$$

This allows us to use a simple recurrent neural network for modeling $q_\phi$. Also, to compute $p_\theta(\mathbf{Z}_t|\mathcal{X}, \mathbf{A}_{<t}, \mathbf{Z}_{<t})$ using a recurrent neural network, we simplify it by using a surrogate term $p_\theta(\mathbf{Z}_t|\mathbf{X}_{\leq t}, \mathbf{A}_{<t}, \mathbf{Z}_{<t})$. Applying the above substitutions into Eq. equation 4 gives:

$$
\begin{aligned}
\mathcal{L}^{sVFE}(\theta, \phi) = & -\sum_{t=0}^{T} \Big[ \mathbb{E}_{q_\phi(Z_{\leq t}|X_{\leq t}, A_{\leq t}, Z_{<t})} \big[ \log p_\theta(\mathbf{A}_t|\mathbf{Z}_t) \big] \\
& + \mathcal{D}^{KL} \big[ q_\phi(\mathbf{Z}_t|\mathbf{X}_{\leq t}, \mathbf{A}_{\leq t}, \mathbf{Z}_{<t}) \| p_\theta(\mathbf{Z}_t|\mathbf{X}_{\leq t}, \mathbf{A}_{<t}, \mathbf{Z}_{<t}) \big] \Big].
\end{aligned}
\tag{6}
$$

The conditional probabilities in Eq. equation 6 capture the inherent causal structure and temporal coherence of the temporal spiking activity networks. This sVFE underpins the TAVRNN framework.

### 4.2.2 RECURRENT GRAPH NEURAL NETWORK

Here, we describe a model parameterization using a graph RNN for the sVFE Eq. 6. Initially, the conditional latent prior and approximate posterior in Eq. 6 are assumed to follow Gaussian

distributions:

$$p_\theta(\mathbf{Z}_t|\mathbf{X}_{<t}, \mathbf{A}_{<t}, \mathbf{Z}_{<t}) = \mathcal{N}(\boldsymbol{\mu}_t^{\text{prior}}, \boldsymbol{\Sigma}_t^{\text{prior}}) \tag{7a}$$

$$q_\phi(\mathbf{Z}_t|\mathbf{X}_{\leq t}, \mathbf{A}_{\leq t}, \mathbf{Z}_{<t}) = \mathcal{N}(\boldsymbol{\mu}_t^{\text{enc}}, \boldsymbol{\Sigma}_t^{\text{enc}}), \tag{7b}$$

with isotropic covariances $\boldsymbol{\Sigma}_t^{\text{prior}} = \text{Diag}(\sigma_t^{\text{prior}^2})$, $\boldsymbol{\Sigma}_t^{\text{enc}} = \text{Diag}(\sigma_t^{\text{enc}^2})$, and $\text{Diag}(\cdot)$ denoting the diagonal function. To enable gradient descent optimization of the sVFE (Eq. 6), the pairs of mean and standard deviation in Eq. 7 are modeled as:

$$(\boldsymbol{\mu}_t^{\text{prior}}, \boldsymbol{\Sigma}_t^{\text{prior}}) = \varphi_\theta^{\text{prior}}(\mathbf{H}_{t-1}) \tag{8a}$$

$$(\boldsymbol{\mu}_t^{\text{enc}}, \boldsymbol{\Sigma}_t^{\text{enc}}) = \Phi_\phi^{\text{enc}}(\varphi_\theta^{\text{x}}(\mathbf{X}_t), \mathbf{H}_{t-1}, \mathbf{A}_t). \tag{8b}$$

In this configuration, the prior model $\varphi_\theta^{\text{prior}}$, the measurement feature model $\varphi_\theta^{\text{x}}$, and the state feature model $\varphi_\theta^{\text{z}}$ are designed as fully connected neural networks. Meanwhile, the encoder model $\Phi_\phi^{\text{enc}}$ is implemented as a GNN. The memory-embedding recurrent states $H_t$ in Eq. 8 are derived as follows:

$$\mathbf{H}_t = \Phi_\theta^{\text{rnn}}(\varphi_\theta^{\text{x}}(\mathbf{X}_t), \varphi_\theta^{\text{z}}(\mathbf{Z}_t), \mathbf{H}_{t-1}, \mathbf{A}_t), \tag{9}$$

where the recurrent model $\Phi_\theta^{\text{rnn}}$ is implemented as a spatial-aware Gated Recurrent Unit (GRU). According to Eq. 9, $\mathbf{H}_t$ functions as the memory embeddings for the historical path $\mathbf{Z}_{\leq t}, \mathbf{X}_{<t}, \mathbf{A}_{<t}$. Subsequently, the likelihood of the adjacency matrix in Eq. 2 is modeled as a Bernoulli distribution:

$$p_\theta(\mathbf{A}_t|\mathbf{Z}_t) = \text{Bernoulli}(\hat{\mathbf{A}}_t), \tag{10}$$

where $\hat{\mathbf{A}}_t$ is the reconstructed adjacency matrix, derived using a matrix product followed by sigmoid activation:

$$\hat{\mathbf{A}}_t = \sigma(\mathbf{Z}_t \times \mathbf{Z}_t^T). \tag{11}$$

In summary, the end-to-end integration of the prior (Eq. 8a), encoder (Eq. 8b), recurrent module (Eq. 9), and inner-product decoder (Eq. 11) forms a probabilistic recurrent graph autoencoder. This model first constructs sequential stochastic hierarchical latent embedding spaces on $\{\mathbf{Z}_t, \mathbf{H}_t\}_{t=0}^T$ and then utilizes these embeddings to perform stochastic estimation of the adjacency matrices $\{\hat{\mathbf{A}}_t\}_{t=0}^T$. By optimizing the sVFE (Eq. 6) with respect to the model parameters $\{\theta, \phi\}$, these embedding spaces adapt to capture a wide array of stochastic spatiotemporal variations across dynamic networks in an entirely unsupervised manner. Further details of the method are provided in Appendix A.7 and A.9.

### 4.2.3 TEMPORAL ATTENTION-BASED MESSAGE PASSING AND SPATIALLY-AWARE GRU

To more accurately reflect spatiotemporal dependencies, we reparameterized the recurrent model (Eq. 9) to include a spatially-aware GRU. This modification facilitates dynamic updates of the recurrent states over time. The update gate $S_t$, reset gate $R_t$, and candidate activation $\tilde{\mathbf{H}}_t$ are calculated as:

$$\mathbf{S}_t = \sigma(\Phi_{xz}(\mathbf{X}, \mathbf{A}_t) + \Phi_{hz}(\mathbf{H}_{t-1}, \mathbf{A}_t)) \tag{12}$$

$$R_t = \sigma(\Phi_{xr}(\mathbf{X}, \mathbf{A}_t) + \Phi_{hr}(\mathbf{H}_{t-1}, \mathbf{A}_t)) \tag{13}$$

$$\tilde{\mathbf{H}}_t = \tanh(\Phi_{xh}(\mathbf{X}, \mathbf{A}_t) + \Phi_{hh}(R_t \odot \mathbf{H}_{t-1}, \mathbf{A}_t)) \tag{14}$$

Finally, the output of the GRU will be computed as:

$$\hat{\mathbf{H}}_t = \mathbf{S}_t \odot \mathbf{H}_{t-1} + (1 - \mathbf{S}_t) \odot \tilde{\mathbf{H}}_t \tag{15}$$

These equations describe the forward pass of our spatially-aware GRU, improving its capacity to process and incorporate spatial information through time, where $\mathbf{X} = [\varphi_x(\mathbf{X}_t), \varphi_z(\mathbf{Z}_t)]^T$. Although $\hat{\mathbf{H}}_t$ could serve as the final value for $\mathbf{H}_t$, given the temporal nature of our graph data, we consider a global state for the entire graph at each time step. While the GRU adds memory to the states, in our GNN structure, each node's state updates based on local information from its neighbors. For this reason, we add a hypothetical node to the graph which is connected to all other nodes. The state of this node is supposed to represent the global state of the graph. According to the dynamic nature of the graph's state, we let the model compute the final value of $\mathbf{H}_t$ through an attention mechanism on $\hat{\mathbf{H}}_t, \mathbf{H}_{t-1}, \mathbf{H}_{t-2}, \ldots$ and $\mathbf{H}_{t-w}$ (see Fig. 2). Mathematical details of this temporal attention module are presented in Appendix A.8. Using the above equations, $\mathbf{H}_t$ serves as memory embeddings that capture graph-structured temporal information from previous latent state sequences. This model replaces the conventional GRU's FCNNs with single-layer GNNs $\{\Phi_{xz}, \Phi_{hz}, \Phi_{xr}, \Phi_{hr}, \Phi_{xh}, \Phi_{hh}\}$ that incorporate a message passing scheme. This adaptation enables the GRU to efficiently leverage both the spatial topologies and temporal dependencies in dynamic graph data.

### 4.3 BASELINES

We used the following unsupervised node-level embedding methods as baselines since our datasets and study focus on unlabeled node sets (see Appendix A.10): 1) **VGAE** (31): Unsupervised framework using a variational auto-encoder with a graph convolutional network encoder and an inner product decoder. 2) **DynGEM** (20): Deep auto-encoder model to generate node embeddings at each time snapshot $t$, initialized from the embedding at $t - 1$. 3) **DynAE** (32): Autoencoder model using multiple fully connected layers for both encoder and decoder to capture highly non-linear interactions between nodes at each time step and across multiple time steps. 4) **DynRNN** (32): RNN-based model using LSTM networks as both encoder and decoder to capture long-term dependencies in dynamic graphs. 5) **DynAERNN** (32): Employs a fully connected encoder to acquire low-dimensional hidden representations, passed through an LSTM network and a fully connected decoder. 6) **GraphERT** (33): Leverages graph embedding representation using transformers with a masked language model on sequences of graph random walks.

## 5 RESULTS

We first evaluate all methods on a classification task using the rat hippocampal dataset, where the ground truth labels are available and correspond to the rat's position on the track. After demonstrating TAVRNN's competitiveness with state-of-the-art temporal graph embedding methods, we proceed to the *DishBrain* dataset for a clustering task. In this setting, characterized by higher dimensionality and intricate single-unit dynamics across varying game performance levels, TAVRNN proves its strength, significantly outperforming all baseline methods.

### 5.1 RAT HIPPOCAMPUS DATASET

Table 1 presents a comparison of the TAVRNN model and baseline methods in the classification task using the rat hippocampal dataset across multiple evaluation metrics. Among the methods, only GraphERT achieved a higher accuracy than TAVRNN, although TAVRNN closely approached its performance and surpassed GraphERT in terms of recall.

Table 1: Comparison of classification performance on rat hippocampal data.

| Method | Accuracy (%) | Recall (%) | Precision (%) | F1-Score (%) |
|---|---|---|---|---|
| VGAE | 64.71 ± 12.89 | 77.78 ± 3.08 | 71.91 ± 2.68 | 74.73 ± 7.31 |
| DynGEM | 62.35 ± 12.11 | 77.50 ± 18.43 | 62.72 ± 12.72 | 69.33 ± 10.30 |
| DynAE | 56.47 ± 10.80 | 51.67 ± 11.30 | 59.29 ± 12.14 | 54.30 ± 8.71 |
| DynRNN | 57.65 ± 11.41 | 68.89 ± 27.58 | 67.39 ± 21.59 | 68.13 ± 9.54 |
| DynAERNN | 70.59 ± 13.92 | 77.78 ± 11.31 | 76.52 ± 18.34 | 77.14 ± 11.26 |
| GraphERT | **93.91 ± 2.48** | 94.27 ± 3.46 | **94.31 ± 2.56** | **94.39 ± 2.27** |
| **TAVRNN** | 91.76 ± 6.80 | **94.56 ± 4.80** | 88.56 ± 10.97 | 91.46 ± 6.30 |

Next, we performed an ablation test, by using four additional variations of our proposed model to test if adding each structure helps the downstream task. The results in Table 2 outline that removing Temporal Attention, replacing the Spatial-aware GRU with a conventional GRU, or replacing the Variational Graph Autoencoder with a simpler Graph Autoencoder all lead to significant performance drops across all evaluation metrics for TAVRNN.

Table 2: Ablation study of the proposed TAVRNN framework.

| Model Specification | Accuracy (%) | Recall (%) | Precision (%) | F1-Score (%) |
|---|---|---|---|---|
| Graph Autoencoder + Conventional GRU | 74.12 ± 10.26 | 86.39 ± 12.92 | 75.93 ± 19.95 | 77.72 ± 6.72 |
| Graph Autoencoder + Spatial-aware GRU | 84.71 ± 12.11 | 86.39 ± 10.84 | 84.59 ± 15.26 | 85.47 ± 10.71 |
| Graph Autoencoder + Spatial-aware GRU + Temporal Attention | 87.06 ± 12.00 | 91.73 ± 8.31 | 87.22 ± 15.28 | 88.10 ± 10.06 |
| Variational Graph Autoencoder + Spatial-aware GRU | 88.24 ± 4.32 | 90.83 ± 5.41 | 87.78 ± 9.63 | 88.72 ± 7.72 |
| Variational Graph Autoencoder + Spatial-aware GRU + Temporal Attention | **91.76 ± 6.80** | **94.56 ± 4.80** | **88.56 ± 10.97** | **91.46 ± 6.30** |

### 5.2 TIME COMPLEXITY ANALYSIS

We also analyzed the time complexity of all baseline methods and compared them to TAVRNN. Table 3 provides the order of time complexity for one forward pass on all the $n$ cells for one time window in all methods. In this table, $h_{max}$ stands for the maximum dimensionality of the hidden layers in different algorithms. See Appendix A.10 for more details on how the time complexities

are computed and meaning of various symbols in the Table. As demonstrated in Table 3, all the methods except GraphERT have similar orders of time complexities, but different constant coefficients. Fig. 3 shows the log-log plot of these time complexities against the number of nodes using all the coefficients and hyper parameters as reported in the original paper for each algorithm. It shows that TAVRNN and VGAE exhibit the lowest time complexity, making them the most computationally efficient methods. In contrast, GraphERT shows the highest complexity, leading to a significant increase in run time as the number of nodes in the input graph grows. This large time complexity is consistent with many constant coefficients we see for GraphERT in Table 3.

## 5.3 DISHBRAIN DATASET

Next, we move to test TAVRNN performance on the *DishBrain* dataset. Fig. 4a-b shows the connectivity networks for the top and bottom three time windows across all trials for a sample culture, ranked by hit/miss ratio during both Gameplay and Rest. The heatmaps display pairwise Pearson correlations between channels for each window. The nodes in these heatmaps are sorted by channel type on the HD-MEA, belonging to $Sens$, $L_{up}$, $R_{up}$, $L_{down}$, or $R_{down}$ regions. Across all recorded cultures, Gameplay sessions showed higher average weight, lower modularity, and lower

Table 3: One forward pass time complexity for one time window.

| Method | Complexity |
|---|---|
| VGAE | $\mathcal{O}\left(n \cdot \sum_{i=1}^{k} h_{i-1} \cdot h_i\right) \in \mathcal{O}\left(n^2 \cdot h_{\max}\right)$ |
| DynGEM | $\mathcal{O}\left(n \cdot \sum_{i=1}^{k+1} h_{i-1} \cdot h_i\right) \in \mathcal{O}\left(n^2 \cdot h_{\max}\right)$ |
| DynAE | $\mathcal{O}\left(n \cdot \sum_{i=1}^{k+1} h_{i-1} \cdot h_i\right) \in \mathcal{O}\left(n^2 \cdot h_{\max}\right)$ |
| DynRNN | $\mathcal{O}\left(n \cdot \sum_{i=1}^{k+1} h_{i-1_{LSTM}} \cdot h_{i_{LSTM}}\right) \in \mathcal{O}\left(n^2 \cdot h_{\max}\right)$ |
| DynAERNN | $\mathcal{O}\left(n \cdot \sum_{i=1}^{k} h_{i-1} \cdot h_i + h_{i-1_{LSTM}} \cdot h_{i_{LSTM}}\right) \in \mathcal{O}\left(n^2 \cdot h_{\max}\right)$ |
| GraphERT | $\mathcal{O}\left((\gamma \cdot |p| \cdot |q| \cdot H \cdot k) \cdot n \cdot L^2 \cdot h_{\max}\right) \in \mathcal{O}\left(n \cdot L^2 \cdot h_{\max}\right)$ |
| TAVRNN | $\mathcal{O}\left(n^2 \cdot h_{\max} + n \cdot w \cdot h_{\max}\right)$ |

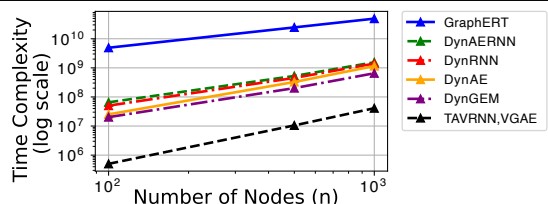

Figure 3: Time complexity of all methods on a log-log plot.

clustering coefficients compared to Rest. Fig. 4c compares these metrics for the best and worst time windows in both Gameplay and Rest, revealing significant differences between the two states but no significant difference between $High^1$ and $Low^1$ during Gameplay. Fig. 4d shows the evolution of these metrics with increasing hit/miss ratio during Gameplay sessions across all recordings. Fig.

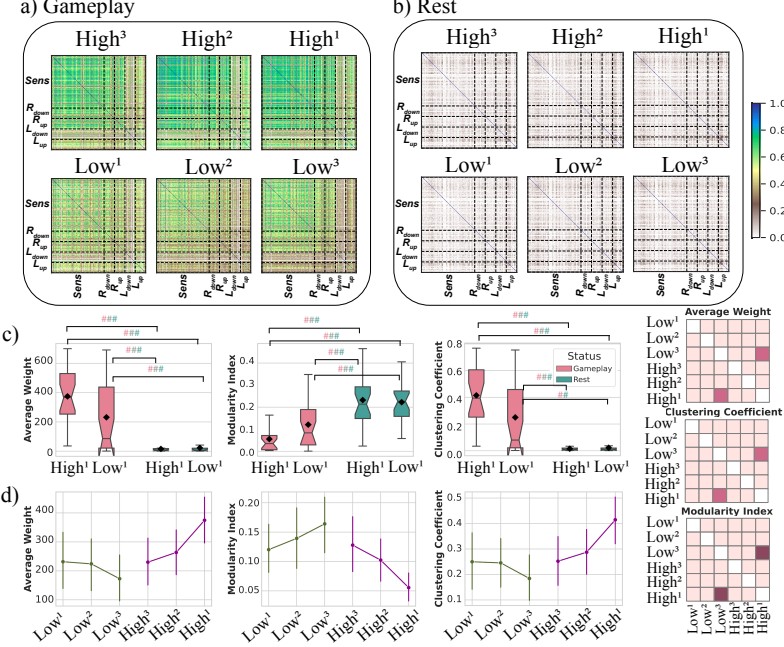

Figure 4: Functional connectivity networks for $High^{1,2,3}$ and $Low^{1,2,3}$ windows in aggregated trials of **a)** Gameplay and **b)** Rest for a sample culture. Average weight, modularity index, and clustering coefficient for **c)** $High^1$ and $Low^1$ across all sessions. Error bands = 1 SE. $\#\#\#p < 0.001$, $\#\#p < 0.01$. **d)** Same metrics for $High^{1,2,3}$ and $Low^{1,2,3}$ in Gameplay across all recordings. Error bars = 95% confidence intervals. **e)** Pairwise Games-Howell post-hoc test between groups.

5a-b visualizes the embeddings for the same sample networks from Fig. 4 using all methods. Nodes are color-coded by their subregions on the HD-MEA. TAVRNN reveals that during high game performance, nodes from different subregions (e.g., $Sens$ or motor subregions for $Up$ and $Down$ movements) form distinct clusters. The clusters become increasingly distinct as game performance reaches its highest level ($High^1$). Notably, the $Sens$ cluster overlaps with motor clusters even at peak performance, suggesting co-activation of a subgroup of $Sens$ cluster with each motor region. This clustering was not detected in the functional connectivity networks of the spiking activity (see for example Fig. 4) but does accord with previous electrophysiological analysis (6). TAVRNN outperforms the other baselines in separating the clusters based on the corresponding channel's subregion label due to its ability to incorporate temporal history of network activity. Additionally, TAVRNN' attention layer enhances its effectiveness. This layer assesses the relevance of historical network activities by comparing their functional connectivity with the current snapshot, thereby significantly influencing the representation in the embedding space and leading to improved performance over the rest. This demonstrates that successful adaptive learning requires synchronous activity between subregions, even as the modularity index of functional connectivity networks decreases during better performance. Our findings uncover the latent topology of the temporal networks revealing that clustering of subregions during successful behavior, as seen in the embedding space, highlights functional modules co-activated during optimal performance, which are not necessarily spatially proximate (see Fig. 1c). Absence of such clustering during poor performance or Rest (see Fig. S4 for these results) implies a disruption in the coordinated activity of these modules suggesting that adaptive learning involves dynamic reorganization of neuronal circuits to optimize behavior.

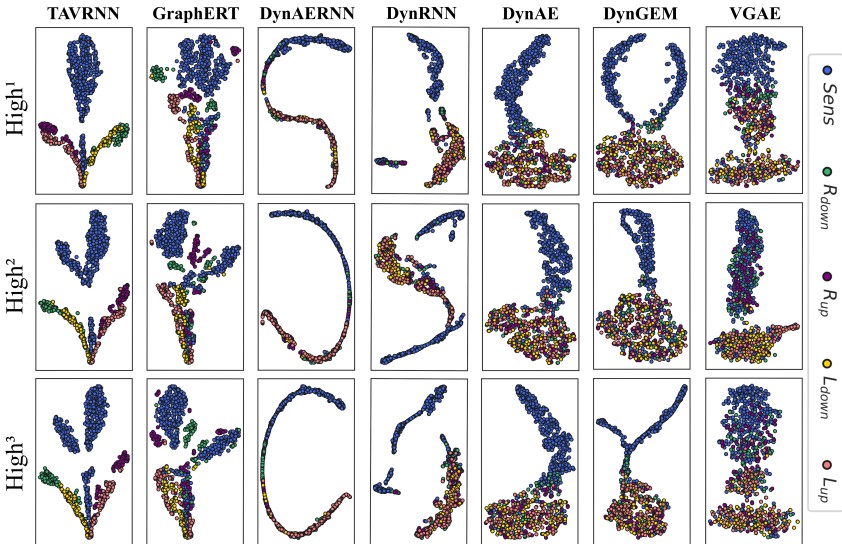

Figure 5: t-SNE visualization of the channels in the embedding space for $High^{1,2,3}$ windows of Gameplay using TAVRNN and all baseline methods for aggregated trials of a sample culture. Each channel is color-coded based on the predefined subregion it belongs to as shown in Fig. 1c. Results from additional cultures, Rest sessions, and $Low^{1,2,3}$ windows are represented in Appendix A.4.

Table 4 represents the comparison results during the best performing Gameplay session ($High^1$) across all cultures in terms of the Silhouette, Adjusted Rand Index (ARI), Homogeneity, and Completeness scores on the clustering task where channels are labeled based on their role ($Sens$, $Up$, or $Down$). We found that TAVRNN outperforms all baseline methods on all metrics. The Silhouette score, which assesses the degree of separation among clusters, indicated some overlap in $High^1$ sessions. This suggests that a complete separation of clusters may not be optimal for the transmission of information between sensory and motor subregions, reflecting a functional co-activation required among channels within these clusters for goal-directed tasks. The ARI evaluated the alignment between true and predicted labels where even TAVRNN showed deviations from perfect alignment, highlighting the challenges of predefined neuron classifications in the *DishBrain* platform. This discrepancy stems from the absence of a definitive ground truth for defining motor subregions, complicating accurate neuron segregation. Notably, the *DishBrain* platform was originally designed

Table 4: Clustering scores on the best ($High^1$) performing windows over all Gameplay sessions.

| Method | Silhouette | ARI | Homogeneity | Completeness |
|---|---|---|---|---|
| VGAE | 0.5385 ± 0.0337 | - 0.0014 ± 0.0004 | 0.0307 ± 0.0012 | 0.0218 ± 0.0006 |
| DynGEM | 0.4220 ± 0.0354 | 0.0035 ± 0.0056 | 0.0043 ± 0.0041 | 0.0044 ± 0.0041 |
| DynAE | 0.4133 ± 0.0366 | 0.0006 ± 0.0026 | 0.0022 ± 0.0019 | 0.0022 ± 0.0019 |
| DynRNN | 0.5551 ± 0.0270 | 0.0168 ± 0.0143 | 0.0145 ± 0.0107 | 0.0149 ± 0.0110 |
| DynAERNN | 0.6051 ± 0.0121 | 0.1391 ± 0.0365 | 0.1053 ± 0.0312 | 0.1059 ± 0.0415 |
| GraphERT | 0.5513 ± 0.0400 | 0.6277 ± 0.1409 | 0.6046 ± 0.1110 | 0.6261 ± 0.0945 |
| **TAVRNN** | **0.6505 ± 0.0215** | **0.8072 ± 0.0372** | **0.7076 ± 0.0357** | **0.7171 ± 0.0331** |

considering various motor subregion configurations for $Up$ and $Down$ paddle movements, with the final regions selected based on optimal performance in experimental cultures (6). Our results indicate that neurons assigned specific roles based on their subregions did not always align with their expected activity patterns, emphasizing the complexity of predicting neuronal behavior in biological systems. Note that the GraphERT method leverages a representation of the entire graph through the CLS token (33), yielding high accuracy in tasks that rely on global network data, such as the classification in rat hippocampus dataset. However, importantly, while TAVRNN demonstrates comparable performance in that task, it significantly outperforms GraphERT in a task where the dynamics of individual nodes are crucial such as the clustering in *DishBrain* dataset. Where single-unit activity is the focus of representation learning rather than population-level behavior, TAVRNN excels by efficiently capturing the temporal latent dynamics of individual nodes in the graph. Additionally, our method exhibits robust performance across datasets with significantly different sampling frequencies, ranging from 40 Hz to 20 kHz for the rat and *DishBrain* datasets.

Overall, our framework provides a valuable tool to facilitate the optimization of neuronal clusters for specific tasks in simulated environments, enhancing the design and efficacy of future experiments. Homogeneity and completeness metrics revealed that clusters contained neurons from multiple classes and did not group all neurons of a class together, even during optimal performance. This indicates a more distributed and nuanced representation of sensory and motor functions within the neuronal network, blurring the predefined boundaries between regions. Our findings highlight the complex interplay of neuronal activity in clustered environments and emphasize the potential of our framework to enhance the understanding and design of future experiments in neuronal clustering and task-specific roles in both biological and simulated systems.

# 6 CONCLUSIONS

By employing a sophisticated representation learning framework, our approach applies a nonlinear dimensionality reduction technique that preserves critical information from individual neurons over time as a groundbreaking method to explore adpative learning in biological neurons. This is different from previous dimensionality reduction methods that examined the temporal trajectory of the entire population as a whole (3). Our methodology enable dissection of the intricate dynamics between single units that underpin successful and unsuccessful behavioral outcomes of neuronal populations. Notably, our TAVRNN framework successfully identified interpretable attributes that correlate with good and poor performance of live biological neurons in a simulated environment of pong such as in the *DishBrain* system. Our findings suggest that in such a system, adaptive learning is facilitated by the dynamic reorganization of neuronal circuits and co-activation of distinct neuronal clusters, optimizing behavioral responses. Moreover, assessing the understanding of the spatial layout of individual channels on the HD-MEA showed that these co-activations are not confined to spatially adjacent subpopulations. Instead, a more complex pattern of self-organization emerges among neuronal subregions that are spatially distant from each other. This indicates a complex pattern of self-organization among distanced neuronal subpopulations, driven endogenously rather than by exogenous influences. These insights open new avenues for targeting specific neuronal mechanisms in skill acquisition and could inform future interventions aimed at enhancing learning and memory, both in health and clinical settings. This finding not only advances our understanding of neuronal behavior in learning tasks but also challenges existing paradigms about the spatial requirements for neuronal co-activation and learning efficacy. A current limitation of our framework is its reliance on undirected networks of functional connectivity. Future iterations could benefit from incorporating directed networks, which would allow for the differentiation between inhibitory and excitatory relationships among channels by using signed correlation values. Additionally, exploring tasks such as link prediction using our framework also represents a promising direction.

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

## A  APPENDIX / SUPPLEMENTAL MATERIAL

The Python implementation of our proposed framework and baseline methods is available at the following Github Repository.

### A.1  CELL CULTURE

Approximately $10^6$ cells were plated on each Multielectrode Array. Neuronal cells were cultured either from the cortices of E15.5 mouse embryos or differentiated from human induced pluripotent stem cells via a dual SMAD inhibition (DSI) protocol or through a lentivirus-based NGN2 direct differentiation protocols as previously described (6). Cells were cultured until plating. For primary mouse neurons, this occurred at day-in-vitro (DIV) 0, for DSI cultures this occurred at between DIV 30 - 33 depending on culture development, for NGN2 cultures this occurred at DIV 3.

### A.2  MEA SETUP AND PLATING

MaxOne Multielectrode Arrays (MEA; Maxwell Biosystems, AG, Switzerland) was used and is a high-resolution electrophysiology platform featuring 26,000 platinum electrodes arranged over an 8 mm2. The MaxOne system is based on complementary meta-oxide-semiconductor (CMOS) technology and allows recording from up to 1024 channels. MEAs were coated with either polyethylenimine (PEI) in borate buffer for primary culture cells or Poly-D-Lysine for cells from an iPSC background before being coated with either 10 μg/ml mouse laminin or 10 μg/ml human 521 Laminin (Stemcell Technologies Australia, Melbourne, Australia) respectively to facilitate cell adhesion. Approximately $10^6$ cells were plated on MEA after preparation as per (6). Cells were allowed approximately one hour to adhere to MEA surface before the well was flooded. The day after plating, cell culture media was changed for all culture types to BrainPhys™ Neuronal Medium (Stemcell Technologies Australia, Melbourne, Australia) supplemented with 1% penicillin-streptomycin. Cultures were maintained in a low O2 incubator kept at 5% CO2, 5% O2, 36°C and 80% relative humidity. Every two days, half the media from each well was removed and replaced with free media. Media changes always occurred after all recording sessions.

### A.3  DISHBRAIN PLATFORM AND ELECTRODE CONFIGURATION

The current DishBrain platform is configured as a low-latency, real-time MEA control system with on-line spike detection and recording software. The DishBrain platform provides on-line spike detection and recording configured as a low-latency, real-time MEA control. The DishBrain software runs at 20 kHz and allows recording at an incredibly fine timescale. This setup captured neuronal electrical activity and provided long-term, safe external electrical stimulation through biphasic pulses that elicited action potentials in neurons, as detailed in previous studies (34). There is the option of recording spikes in binary files, and regardless of recording, they are counted throughout 10 milliseconds (200 samples), at which point the game environment is provided with how many spikes are detected in each electrode in each predefined motor region as described below. Based on which motor region the spikes occurred in, they are interpreted as motor activity, moving the 'paddle' up or down in the virtual space. As the ball moves around the play area at a fixed speed and bounces off the edge of the play area and the paddle, the pong game is also updated at every 10ms interval. Once the ball hits the edge of the play area behind the paddle, one rally of pong has come to an end. The game environment will instead determine which type of feedback to apply at the end of the rally: random, silent, or none. Feedback is also provided when the ball contacts the paddle under the standard stimulus condition. A 'stimulation sequencer' module tracks the location of the ball relative to the paddle during each rally and encodes it as stimulation to one of eight stimulation sites. Each time a sample is received from the MEA, the stimulation sequencer is updated 20,000 times a second, and after the previous lot of MEA commands has completed, it constructs a new sequence of MEA commands based on the information it has been configured to transmit based on both place codes and rate codes. The stimulations take the form of a short square bi-phasic pulse that is a positive voltage, then a negative voltage. This pulse sequence is read and applied to the electrode by a Digital to Analog Converter (or DAC) on the MEA. A real-time interactive version of the game visualiser is available at https://spikestream.corticallabs.com/. Alternatively, cells could be recorded at 'Rest' in a Gameplay environment where activity was recorded to move the

paddle but no stimulation was delivered, with corresponding outcomes still recorded. Using this spontaneous activity alone as a baseline, the Gameplay characteristics of a culture were determined. Low level code for interacting with Maxwell API was written in C to minimize processing latencies-so packet processing latency was typically $<50$ $\mu$s. High-level code was written in Python, including configuration setups and general instructions for game settings. A 5 ms spike-to-stim latency was achieved, which was substantially due to MaxOne's inflexible hardware buffering. Fig. S1 illustrates a schematic view of Software components and data flow in the DishBrain closed loop system.

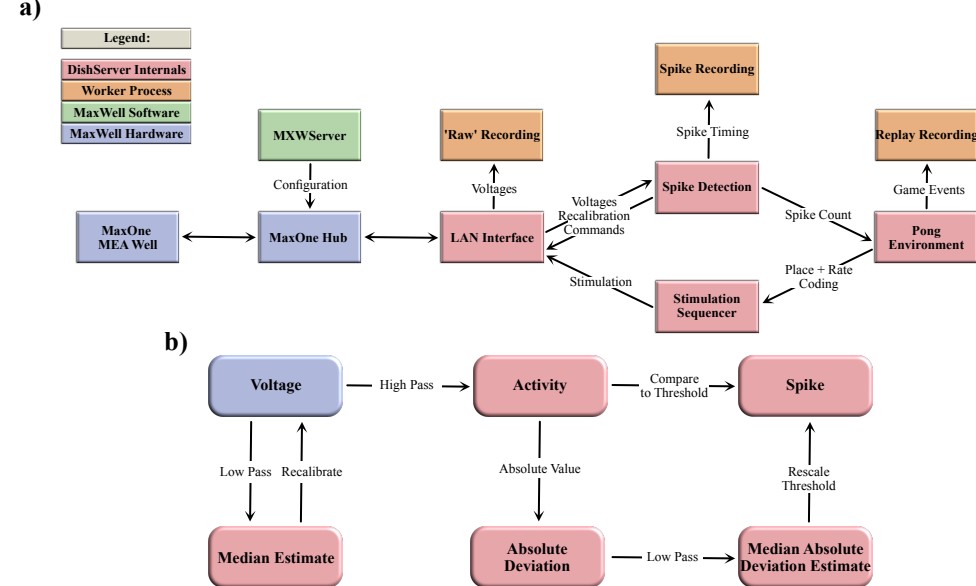

Figure S1: **a, b)** Schematics of software used for DishBrain. **a)** Software components and data flow in the DishBrain closed loop system. Voltage samples flow from the MEA to the 'Pong' environment, and sensory information flows from the 'Pong' environment back to the MEA, forming a closed loop. The blue rectangles mark proprietary pieces of hardware from MaxWell, including the MEA well which may contain a live culture of neurons. The green MXWServer is a piece of software provided by MaxWell which is used to configure the MEA and Hub, using a private API directly over the network. The red rectangles mark components of the 'DishServer' program, a high-performance program consisting of four components designed to run asynchronously, despite being run on a single CPU thread. The 'LAN Interface' component stores network state, for talking to the Hub, and produces arrays of voltage values for processing. Voltage values are passed to the 'Spike Detection' component, which stores feedback values and spike counts, and passes recalibration commands back to the LAN Interface. When the pong environment is ready to run, it updates the state of the paddle based on the spike counts, updates the state of the ball based on its velocity and collision conditions, and reconfigures the stimulation sequencer based on the relative position of the ball and current state of the game. The stimulation sequencer stores and updates indices and countdowns relating to the stimulations it must produce and converts these into commands each time the corresponding countdown reaches zero, which are finally passed back to the LAN Interface, to send to the MEA system, closing the loop. The procedures associated with each component are run one after the other in a simple loop control flow, but the 'Pong' environment only moves forward every 200th update, short-circuiting otherwise. Additionally, up to three worker processes are launched in parallel, depending on which parts of the system need to be recorded. They receive data from the main thread via shared memory and write it to file, allowing the main thread to continue processing data without having to hand control to the operating system and back again. **b)** Numeric operations in the real-time spike detection component of the DishBrain closed loop system, including multiple IIR filters. Running a virtual environment in a closed loop imposes strict performance requirements, and digital signal processing is the main bottleneck of this system, with close to 42 MB of data to process every second. Simple sequences of IIR digital filters is applied to incoming data, storing multiple arrays of 1024 feedback values in between each sample. First, spikes on the incoming data are detected by applying a high pass filter to determine the deviation of the activity, and comparing that to the MAD, which is itself calculated with a subsequent low pass filter. Then, a low pass filter is applied to the original data to determine whether the MEA hardware needs to be re-calibrated, affecting future samples. This system was able to keep up with the incoming data on a single thread of an Intel Core i7-8809G. Figures adapted from (6).

## A.4 Additional Results

In this section, we present the learned representations of the three best performing windows in terms of the culture's hit/miss ratios during Gameplay for two additional cultures in Figs. S2 and S3. The figures repeatedly demonstrate TAVRNN's outperformance over the other baseline methods in identifying clusters of channels that belong to the same region on the HD-MEA.

Additionally, Fig. S4 represents t-SNE visualization of the learned representations of the three best and three worst windows based on hit/miss ratios ($High^{1,2,3}$ and $Low^{1,2,3}$) during Gameplay and Rest, as modeled by TAVRNN for all aggregated trials of an additional sample culture. These visualizations reveal an absence of distinguishable clusters during the rest state or during low-performing periods of gameplay. However, as we progress to time windows associated with higher performance levels in the game, distinct clustering patterns emerge. Notably, channels from the motor regions associated with $Up$ and $Down$ movements form distinct, cohesive clusters, despite the spatial separation of these channels (within each of the $Up$ or $Down$ subregions) on the HD-MEA. Similarly, channels from the $Sens$ region group together into a separate cluster.

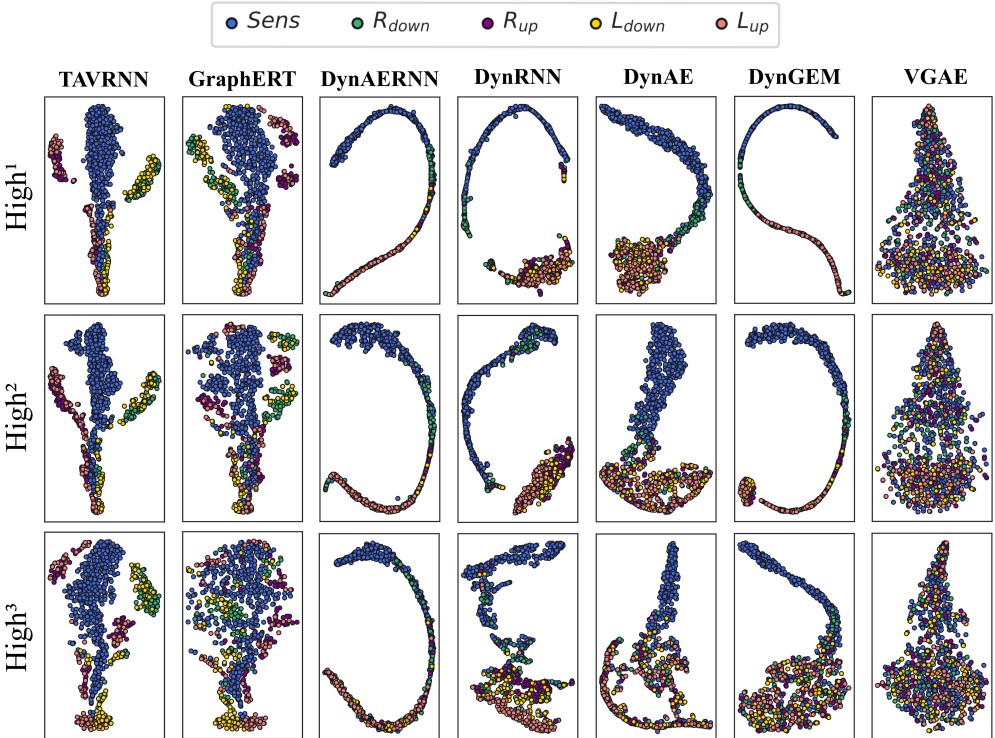

Figure S2: t-SNE visualization of the channels in the embedding space for $High^{1,2,3}$ windows of Gameplay using TAVRNN and all baseline methods for aggregated trials of an additional sample culture. Each channel is color-coded based on the predefined subregion it belongs to as shown in Fig. 1c.

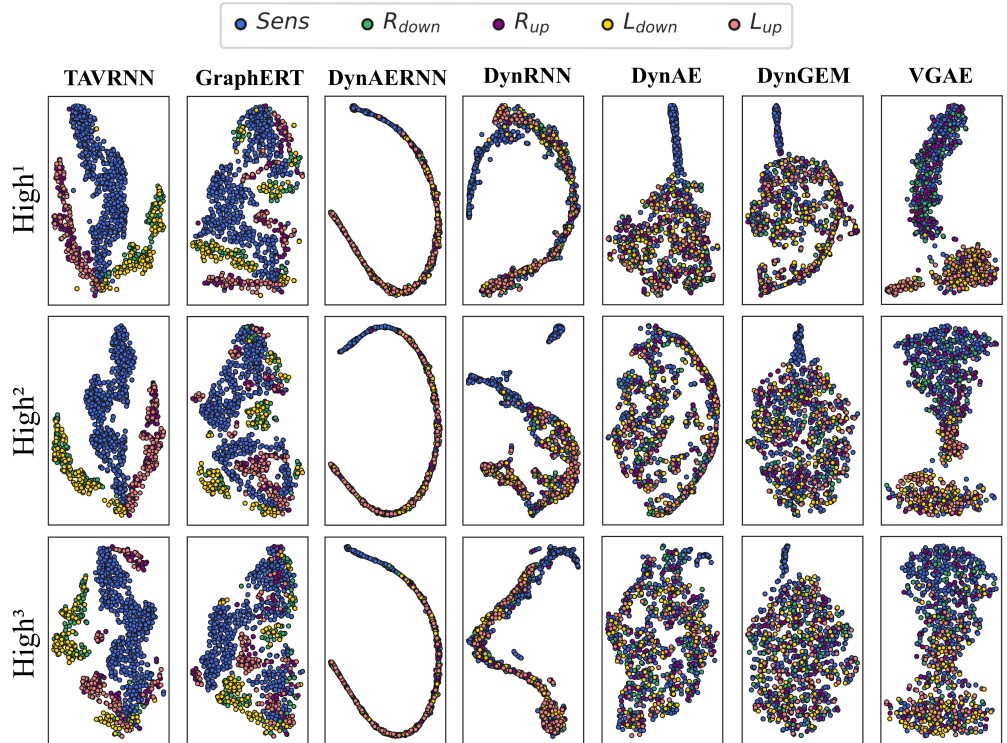

Figure S3: t-SNE visualization of the channels in the embedding space for $High^{1,2,3}$ windows of Gameplay using TAVRNN and all baseline methods for aggregated trials of another sample culture. Each channel is color-coded based on the predefined subregion it belongs to as shown in Fig. 1c.

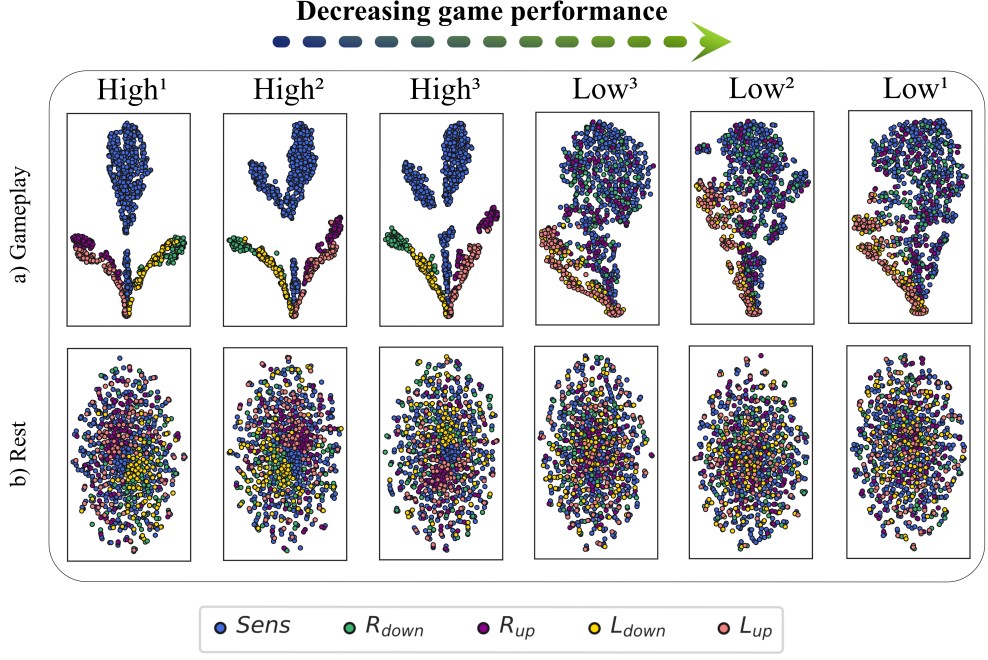

Figure S4: t-SNE visualization of the channels in the embedding space using TAVRNN during the top and bottom three windows ($High^{1,2,3}$ and $Low^{1,2,3}$) in terms of hit-miss-ratio during Gameplay and Rest for aggregated trials of a sample culture. Each channel is color-coded based on the predefined subregion it belongs to as shown in Fig. 1c.

## A.5 CONNECTIVITY INFERENCE MECHANISMS

Methods for inferring connectivity are broadly categorized into two types: *model-free* and *model-based* approaches. Model-free methods rely on descriptive statistics and do not presuppose any specific underlying data generation mechanism, making them versatile for initial analyses. In contrast, model-based methods involve hypothesizing a mathematical model to elucidate the underlying biological processes by estimating its parameters and structure. Typically, these methods analyze time-series data, such as spike trains from individual neurons. However, recent advances have enabled studies to integrate spike inference with connectivity analysis directly from time-series data (35). In this work, we focus on utilizing the model-free methods.

Model-free methods do not presuppose any specific mechanisms underlying the observed data, offering a simpler alternative to model-based approaches. However, these methods do not facilitate the generation of activity data crucial for model validation or predictive analysis. Model-free techniques are primarily divided into two categories: those employing descriptive statistics such as Pearson correlation coefficient (PC) and cross-correlation (CC) and those utilizing information-theoretic measures such as Mutual information (MI), and Transfer entropy (TE) (35; 36; 37; 38; 39; 40; 41).

### A.5.1 GRAPH KERNELS

In light of the diversity of connectivity inference methods discussed previously, each method can generate distinct graph representations from identical datasets. To extract meaningful insights from these varied representations, it is essential to employ a comparison methodology. However, graph comparison is computationally challenging. Ideally, one would verify if two graphs are exactly identical, a problem known as graph isomorphism, which is NP-complete (42). This complexity renders the task computationally prohibitive for large graphs.

To circumvent these difficulties, kernel methods offer a viable alternative. Kernels are functions designed to measure the similarity between pairs, enabling the transformation of objects into a high-dimensional space conducive to linear analysis methods. Graph kernels, specifically, facilitate the comparison of graphs by evaluating their structure, topology, and other attributes, thus proving instrumental in machine learning applications for graph data, such as clustering and classification (43; 44; 45).

Graph kernels vary in their approach to measuring similarity. Some rely on neighborhood aggregation, which consolidates information from adjacent nodes to form local feature vectors (46; 47; 48), while others utilize assignment and matching techniques to establish correspondences between nodes in different graphs (49). Additionally, some kernels identify and compare subgraph patterns (50), and others analyze walks and paths to capture structural nuances (51).

Here we concentrate on neighborhood aggregation methods, particularly pertinent for analyzing connectivity graphs derived from neuronal recordings, typically involving fewer than 1000 nodes without definitive node labels. These methods are also foundational for the graph neural network models. We exemplify this approach with the 1-dimensional *Weisfeiler-Lehman* (1-WL) algorithm (46), illustrating its application and effectiveness.

***Weisfeiler-Lehman* Algorithm** The *Weisfeiler-Lehman* (WL) graph kernel is a sophisticated approach for computing graph similarities, which leverages an iterative relabeling scheme based on the *Weisfeiler-Lehman* isomorphism test. This method extends the basic graph kernel framework by incorporating local neighborhood information into the graph representation, making it particularly effective for graph classification tasks.

Consider a graph $G = (V, E, \ell)$, where $V$ is the set of vertices, $E$ is the set of edges, and $\ell : V \to \Sigma$ is a labeling function that maps each vertex to a label from a finite alphabet $\Sigma$. Initially, each vertex is assigned a label based on its original label or degree.

Define $\ell^0 = \ell$. At each iteration $i$, a new labeling $\ell^i$ is computed as follows:

$$\ell^{i+1}(v) = \text{HASH}\left(\ell^i(v), \{\!\{\ell^i(u) \mid u \in N(v)\}\!\}\right)$$

where $N(v)$ denotes the set of neighbors of vertex $v$ and $\{\!\{\cdot\}\!\}$ denotes a multiset, ensuring that the labels of neighboring vertices are considered without regard to their order. The function HASH maps the concatenated labels to a new, unique label. The algorithm continues iteratively, relabeling vertices until the labels converge or no new labels are produced (Fig. S5).

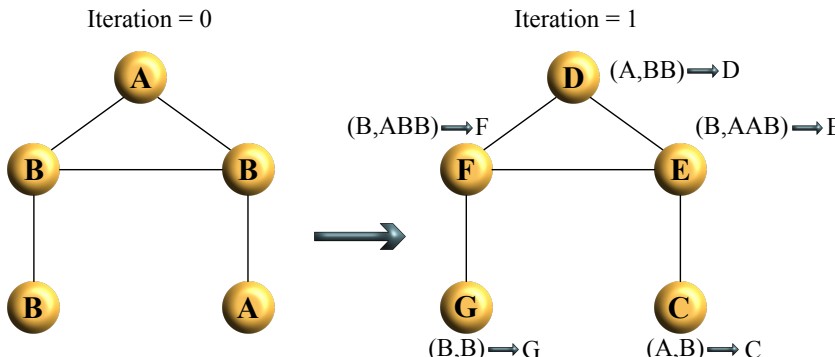

Figure S5: Illustration of the 1-dimensional *Weisfeiler-Lehman* (1-WL) algorithm. This diagram demonstrates how the 1-WL algorithm initially encounters overlapping node labels and, through one iteration, assigns unique labels to each node based on their positions within the graph.

After each iteration $i$, compute a feature vector $\phi^i(G)$ as the histogram of the labels across all vertices:

$$\phi^i(G) = \left(\#\{v \in V \mid \ell^i(v) = k\}\right)_{k \in \mathcal{K}}$$

where $\mathcal{K}$ is the set of all possible labels at iteration $i$.

The WL kernel between two graphs $G$ and $G'$ is defined as the sum of base kernel evaluations on the corresponding histograms at each iteration:

$$K(G, G') = \sum_{i=0}^{h} K_{\text{base}}\left(\phi^i(G), \phi^i(G')\right)$$

where $K_{\text{base}}$ is typically chosen to be the linear kernel $K_{\text{base}}(\phi, \phi') = \phi \cdot \phi'$, and $h$ is a predefined number of iterations, determining the depth of neighborhood aggregation.

In this study, we analyzed 437 recording sessions, comprising 262 Gameplay and 175 Rest sessions, to construct functional connectivity graphs. These graphs were derived using four distinct network inference algorithms: Zero-lag Pearson Correlations (PC), Cross-Correlation (CC), Mutual Information (MI), and Transfer Entropy (TE). For the PC analysis, connectivity matrices were thresholded at varying levels $t \in \{0, 20, 40, 60, 80\}\%$, retaining only the strongest connections as determined by their absolute correlation values. For both CC and TE, we explored delay values $d \in \{1, 2, 3, 4\}$. Each method produced 437 distinct networks.

Subsequently, a Weisfeiler-Lehman (WL) graph kernel with depth $h = 4$ was utilized to compute the kernel matrix $\mathbf{K}$, which was then employed in a Support Vector Machine (SVM) classifier to distinguish between Gameplay and Rest sessions. Classification effectiveness was evaluated through a 5-fold cross-validation on the *DishBrain* dataset, achieving the results summarized in Table S1. Notably, classification performance for CC and TE improved with increasing delay values, reflecting enhanced discriminative power of the graph kernels with longer embedding lengths. However, this increase in delay also introduced greater computational complexity, presenting challenges in scalability and traceability.

### A.6 Marchenko-Pastur Distribution and Shuffling Procedure

In random matrix theory, the Marchenko-Pastur (MP) distribution describes the asymptotic behavior of the eigenvalues of large-dimensional sample covariance matrices. Consider a random matrix $\mathbf{A} \in \mathbb{R}^{p \times n}$, where $p$ represents the number of variables (e.g., neurons or channels) and $n$ represents the number of observations (e.g., time points). The sample covariance matrix is defined as:

$$\mathbf{C} = \frac{1}{n}\mathbf{A}^T\mathbf{A}$$

As both $p$ and $n$ grow large, while the ratio $\eta = \frac{p}{n}$ remains constant, the empirical distribution of the eigenvalues of $\mathbf{C}$ converges to the Marchenko-Pastur distribution (28):

Table S1: Network inference method performance on *DishBrain* dataset

| Network inference method | Avg. accuracy | Std. dev. |
|---|---|---|
| PC (t = 0%) | 0.672 | 0.062 |
| PC (t = 20%) | 0.735 | 0.073 |
| PC (t = 40%) | **0.831** | **0.034** |
| PC (t = 60%) | 0.552 | 0.019 |
| PC (t = 80%) | 0.464 | 0.047 |
| CC (d=1) | 0.432 | 0.126 |
| CC (d=2) | 0.546 | 0.082 |
| CC (d=3) | 0.698 | 0.092 |
| CC (d=4) | 0.763 | 0.103 |
| MI | 0.722 | 0.057 |
| TE (d=1) | 0.657 | 0.073 |
| TE (d=2) | 0.688 | 0.112 |
| TE (d=3) | 0.731 | 0.028 |
| TE (d=4) | 0.794 | 0.063 |

$$\rho(\lambda) = \frac{\sqrt{(\lambda_+ - \lambda)(\lambda - \lambda_-)}}{2\pi\sigma^2\lambda\eta}$$

for $\lambda \in [\lambda_-, \lambda_+]$, where $\sigma$ is the variance of the entries of matrix $\mathbf{A}$ and:

$$\lambda_\pm = \sigma^2 \left(1 \pm \sqrt{\eta}\right)^2$$

In the case where $\eta > 1$, which holds for our data ($p$ is large relative to $n$), the MP distribution suggests that most of the eigenvalues will be close to zero. As a result, the sample covariance matrix is likely to be ill-conditioned, and hence unreliable for further analysis.

### A.6.1 SHUFFLING PROCEDURE FOR CORRELATION ANALYSIS

To account for potential spurious correlations due to ill-conditioning of the sample covariance matrix, we perform a shuffling control procedure:

1. **Shuffle Time Points:** The time points of each channel are independently shuffled while maintaining the channel identity. This process destroys any temporal correlation, ensuring that the correlation between channels is not influenced by the original time structure.

2. **Multiple Iterations:** The shuffling procedure is repeated multiple times (e.g., we chose 1000 iterations) to build a null distribution of correlations for each pair of channels.

3. **Confidence Intervals:** Based on the null distribution obtained from the shuffled data, we compute confidence intervals for each pair of channels. Correlation values from the original data that lie outside of the 95% confidence interval are considered statistically significant.

This approach provides a robust method for identifying significant correlations in the presence of potential ill-conditioning of the sample covariance matrix.

### A.7 UNSUPERVISED SEQUENTIAL VFE (SVFE) LOSS

In a Variational Graph Auto Encoder (VGAE), an encoder network is responsible for learning the latent embeddings $\{\mathbf{Z}_t\}_{t=0}^T$, which capture the representation of nodes in a reduced-dimensional space. The probablity of an edge between nodes $i$ and $j$ in the reconstructed graph is determined by the inner product of their respective latent embeddings, $\mathbf{Z}_{t,i}$ and $\mathbf{Z}_{t,j}$. This process is usually accompanied by a sigmoid activation function to constrain the output values between 0 and 1:

$$\hat{a}_{t,ij} = \sigma(\mathbf{Z}_{t,i} \cdot \mathbf{Z}_{t,j}^T). \tag{S1}$$

In this context, $\sigma$ represents the sigmoid function, $\mathbf{Z}_{t,i}$ refers to the $i$th row of the matrix $\mathbf{Z}_t$, and $\hat{a}_{t,ij}$ corresponds to the $(i,j)$th element of the matrix $\hat{\mathbf{A}}_t$, indicating the predicted probability of an edge between nodes $i$ and $j$ at time $t$.

Considering that $\hat{a}_{t,ij}$ indicates the probability of an edge, the likelihood of the observed adjacency matrix $\mathbf{A}_t$ based on the embeddings can be independently modeled for each edge using a Bernoulli distribution:

$$p_\theta(\mathbf{A}_t|\mathbf{Z}_{\leq t}, \mathbf{X}_{<t}, \mathbf{A}_{<t}) = \prod_{i,j=1}^{N} \hat{a}_{t,ij}^{a_{t,ij}}(1 - \hat{a}_{t,ij})^{1-a_{t,ij}}. \tag{S2}$$

In this case, $a_{t,ij}$ represents the actual entry in the adjacency matrix $\mathbf{A}_t$, signifying the presence, absence, or weight (for weighted graphs) of an edge between nodes $i$ and $j$.

The log-likelihood of the adjacency matrix, $\log p_\theta(\mathbf{A}_t|\mathbf{Z}_{\leq t}, \mathbf{X}_{<t}, \mathbf{A}_{<t})$, can be expressed as the negative binary cross entropy (BCE):

$$\mathcal{L}^{\text{BCE}}(\theta, \phi) = \sum_{i,j=1}^{N} \Big[ a_{t,ij} \log \hat{a}_{t,ij} + (1 - a_{t,ij}) \log(1 - \hat{a}_{t,ij}) \Big]. \tag{S3}$$

We approximate the first expectation term in the sequential VFE (sVFE) using Monte Carlo integration as follows:

$$\mathbb{E}_{q_\phi(z_t|x_{\leq t})} \left[ \log p_\theta(\mathbf{A}_t|\mathbf{Z}_{\leq t}, \mathbf{X}_{<t}, \mathbf{A}_{<t}) \right] = \frac{1}{M} \sum_{k=1}^{M} \mathcal{L}^{\text{BCE}}(\mathbf{Z}_t^k). \tag{S4}$$

Here, $k$ represents the particle index, and $M$ refers to the number of particles, which may be set to 1 when the mini-batch size is sufficiently large (52).

Latent particles $\mathbf{Z}_t^k$ are sampled from $q_\phi(\mathbf{Z}_t|\mathbf{X}_{\leq t}, \mathbf{A}_{\leq t}, \mathbf{Z}_{<t})$ as described by Eq. (7b), utilizing the reparameterization trick $\mathbf{Z}_t^k = \mu_t^{\text{enc}} + \sigma_t^{\text{enc}} \odot \epsilon_t^k$, where $\epsilon_t^k$ is drawn from $\mathcal{N}(0, I)$ and $\odot$ represents the Hadamard (element-wise) product. Recurrent state particles $\mathbf{H}_t^k$ are derived using Eq. (9), based on $\mathbf{Z}_{t-1}^k$ and the previous time-step's state $\mathbf{H}_{t-1}^k$.

Additionally, an analytical solution for the Kullback-Leibler divergence $D_{\text{KL}}$ in the sequential VFE Eq. (4) can be derived in closed form as:

$$D_{\text{KL}}(\theta, \phi) = \frac{1}{2} \sum_{i,j=1}^{N,D} \left[ \frac{\sigma_{t,ij}^{\text{enc2}}}{\sigma_{t,ij}^{\text{prior2}}} - \log \frac{\sigma_{t,ij}^{\text{enc2}}}{\sigma_{t,ij}^{\text{prior2}}} + \frac{(\mu_{t,ij}^{\text{enc}} - \mu_{t,ij}^{\text{prior}})^2}{\sigma_{t,ij}^{\text{prior2}}} - 1 \right] \tag{S5}$$

This KLD loss is deterministic, thereby eliminating the need for Monte Carlo approximation. It quantifies the statistical distance between the conditional prior as specified in Eq. (7a) and the approximate posterior in Eq. (7b). Optimizing this measure strengthens the causality within the latent space, as the prior Eq. (8a) focuses on the influence of preceding graphs and embeddings $\{\mathbf{X} < t, \mathbf{A} < t, \mathbf{Z} < t\}$.

By integrating Eq. (S4) and Eq. (S5) into Eq. (4), we formulate an unsupervised sVFE loss that forms the foundation of the proposed TAVRNN framework:

$$\mathcal{L}^{\text{TAVRNN}}(\theta, \phi) = \mathcal{L}^{\text{BCE}}(\theta, \phi) + \mathcal{D}^{\text{KL}}(\theta, \phi)$$

$$= \underbrace{\frac{1}{M} \sum_{t=0}^{T} \sum_{k=1}^{M} \sum_{i,j=1}^{N} \left[ a_{t,ij} \log \sigma \left( \mathbf{Z}_t^k \times \mathbf{Z}_t^{k^T} \right) + (1 - a_{t,ij}) \log \left( 1 - \sigma \left( \mathbf{Z}_t^k \times \mathbf{Z}_t^{k^T} \right) \right) \right]}_{\mathcal{L}^{\text{BCE}}(\theta, \phi)}$$

$$+ \underbrace{\frac{1}{2} \sum_{t=0}^{T} \sum_{i,j=1}^{N} \left[ \frac{(\sigma_{t,ij}^{\text{enc}} + \epsilon)^2}{(\sigma_{t,ij}^{\text{prior}} + \epsilon)^2} - \log \frac{(\sigma_{t,ij}^{\text{enc}} + \epsilon)^2}{(\sigma_{t,ij}^{\text{prior}} + \epsilon)^2} + \frac{(\mu_{t,ij}^{\text{enc}} - \mu_{t,ij}^{\text{prior}})^2}{(\sigma_{t,ij}^{\text{prior}} + \epsilon)^2} - 1 \right]}_{\mathcal{D}^{\text{KL}}(\theta, \phi)}.$$

$$(\text{S6})$$

### A.8 TEMPORAL ATTENTION MECHANISM

The goal of this section is to present the mathematical details of the temporal attention mechanism for computing $\mathbf{H}_t$ for $\hat{\mathbf{H}}_t$ and $\mathbf{H}_{t-1}, \mathbf{H}_{t-2}, \ldots \mathbf{H}_{t-w}$. Let the $d_h$ dimensional row vector $\overline{s}_i$ present the global state of the graph at time step $i$. [1] Also let $\overline{\mathbf{S}}$ be a $(w+1) \times (w+1)$ matrix that its $i$-th row is equal to $\overline{s}_{t-w-1+i}$. We compute the query vector $q$ and the key matrix $K$ as follows:

$$q = \overline{s}_t \times \mathbf{W}_q + b_q \tag{S7}$$

$$\mathbf{K} = \overline{\mathbf{S}} \times \mathbf{W}_k + b_k \tag{S8}$$

Here, the $d_h \times d_k$ matrices $\mathbf{W}_q$ and $\mathbf{W}_k$, and also the $d_k$ dimensional row vectors $b_q$ and $b_k$ are learnable parameters of our model. Then, the attention vector $\alpha$, which is a $w + 1$ dimensional row vector, will be defined as:

$$\alpha = \text{softmax} \left( \frac{q \times K^T}{\sqrt{d_k}} \right). \tag{S9}$$

Let us define the value matrices as follows:

$$\mathbf{V}_i = \mathbf{H}_{t-w-1+i} \times \mathbf{W}_v + b_v \qquad \forall 1 \leq i \leq w, \tag{S10}$$

and

$$\mathbf{V}_{w+1} = \hat{\mathbf{H}}_t \times \mathbf{W}_v + b_v, \tag{S11}$$

where the $d_h \times d_h$ matrix $\mathbf{W}_v$ and the $d_h$ dimensional row vector $b_v$ are the other learnable parameters of our model.

Finally, the state matrix $\mathbf{H}_t$ will be computed as follows:

$$\mathbf{H}_t = \sum_{i=1}^{w} \alpha_i \times \mathbf{V_i}. \tag{S12}$$

### A.9 TAVRNN MODEL TRAINING HYPERPARAMETERS

All the experiments were run on a 2.3 GHz Quad-Core Intel Core i5. PyTorch 1.8.1 was used to build neural network blocks.

We configured our TAVRNN model by employing graph-structured GRU-Attention with a single recurrent hidden layer consisting of 32 units. The window size $w$ in the attention mechanism is set to the maximum possible for every time step, allowing the model to attend to all previous time steps, including the very first one. The functions $\varphi_\theta^x$ and $\varphi_\theta^z$ in Eqs. (8b) and (9) are implemented using a 32-dimensional fully-connected layer. For the function $\varphi_\theta^{\text{prior}}$ in Eq. (8a), we use two 32 and 8 dimensional fully-connected layers. To model $\boldsymbol{\mu}_t^{\text{enc}}$ and $\boldsymbol{\Sigma}_t^{\text{enc}}$ we employ a 2-layer GCN with 32 and 8 layers, respectively. Our model is initialized using Glorot initialization (53). The learning rate for training is set to 0.01. Training is performed over 1000 epochs using the Adam SGD optimizer (54).

The implementation of our proposed model is available at the following Github Repository.

---

[1] For $i < t$, $\overline{s}_i$ is equal to that row of $\mathbf{H}_i$ which corresponds to the hypothetical node that is connected to all other nodes. Also, $\overline{s}_t$ is equal to the corresponding row of $\hat{\mathbf{H}}_t$.

## A.10 Time Complexity Analysis

In this section, we will compute the time complexity for each method. This analysis provides insights into the computational cost and efficiency of different methods for representation learning of temporal graph data. More specifically, we compute the time complexity of a forward pass on the entire set of the graph nodes in one snapshot for each method.

### A.10.1 GraphERT:

GraphERT is a Transformer-based model for temporal graphs. It uses multiple random walks with different transition parameters $p$ and $q$ to capture the neighborhood structure around each node at specific time steps. These random walks are fed into a Transformer, which learns node-to-node interactions and their temporal relevance using multi-head attention.

Random Walks Generation:

For each graph snapshot, the algorithm generates $\gamma \times n \times |p| \times |q|$ random walks, where:

- $\gamma$ is the number of random walks starting from each node for each pair of values assigned to $p$ and $q$.
- $n$ is the number of nodes in the graph.
- $|p|$ and $|q|$ are the number of different values for the hyperparameters $p$ and $q$.

The time complexity for generating the random walks is:

$$\mathcal{O}(\gamma \times n \times |p| \times |q| \times L)$$

where $L$ is the length of each random walk.

Transformer Processing:

Each random walk is processed by the Transformer. The time complexity of the Transformer is dominated by the self-attention mechanism, which scales quadratically with the sequence length and linearly with the number of attention heads.

For each random walk, the time complexity is:

$$\mathcal{O}(L^2 \times h_{\max} \times H \times k)$$

where:

- $L$ is the random walk length.
- $h_{\max}$ is the maximum dimensionality of the representation vectors used in different transformer layers. In the original implementation of GraphERT we have $h_{\max} = d$, but in general it can take any value larger than or equal to $d$.
- $H$ is the number of attention heads.
- $k$ is the number of layers in the Transformer.

Total Time Complexity:

The total number of random walks is $\gamma \times n \times |p| \times |q|$. Combining the time complexity for random walk generation and Transformer processing, the total time complexity for processing a single graph snapshot is:

$$\mathcal{O}\big(n \cdot \gamma \cdot |p| \cdot |q| \cdot (L + L^2 \cdot h_{\max} \cdot H \cdot k)\big) \in \mathcal{O}\big(n \cdot \gamma \cdot |p| \cdot |q| \cdot (L^2 \cdot h_{\max} \cdot H \cdot k)\big)$$

We can assume that $\gamma$, $|p|$, $q$, $H$ and $k$ are constant values, because they can be fixed values, independent of the graph size ($n$) and the intended dimensionality of the final representations ($d$).

Therefore, we can simplify the total complexity as follows:

$$\mathcal{O}\big((\gamma \cdot |p| \cdot |q| \cdot H \cdot k) \cdot n \cdot L^2 \cdot h_{\max}\big) \in \mathcal{O}\big(n \cdot L^2 \cdot h_{\max}\big)$$

However, it is worth noting that the constant value of this running time is large enough to make practical issues in real experiments. That is why GraphERT shows the most time complexity in Figure 3. Look at Table S2 for more details about the used values for the hyperparameters of this method.

| Method | Hyperparameter | Description / Value |
|---|---|---|
| | $p$ (Return parameter) | Bias for random walks to return to previous node $\in [0.25, 0.5, 1, 2, 4]$ |
| | $q$ (In-out parameter) | Bias for random walks to explore outward $\in [0.25, 0.5, 1, 2, 4]$ |
| GraphERT | Random Walk Length ($L$) | Length of each random walk (32) |
| | Number of Random Walks ($\gamma$) | Number of random walks per node (10) |
| | Embedding Dimension ($d$) | Size of node embeddings (8) |
| | Attention Heads ($H$) | Number of attention heads (4) |
| | Transformer Layers ($k$) | Number of Transformer layers (6) |
| | Learning Rate | Learning rate for the Adam optimizer (1e-4) |

Table S2: Hyperparameters for GraphERT

### A.10.2   VGAE:

To compute the time complexity of a Variational Graph Autoencoder (VGAE) with $n$ nodes, $e$ edges, $k$ Graph Convolutional Network (GCN) layers, and hidden dimensions $h_1, h_2, \ldots, h_k$, where the final latent representation dimension is $d$, we need to analyze the time complexity at each layer of the GCN. This will account for both node-wise and edge-wise operations.

Step 1: GCN Layer Operations

A GCN layer applies a linear transformation followed by neighborhood aggregation. The complexity of a single GCN layer is typically determined by:

- **Node-wise operations**: These involve multiplying the node features by a weight matrix. This has a time complexity of $\mathcal{O}(n \cdot h_{\text{in}} \cdot h_{\text{out}})$, where $h_{\text{in}}$ is the input dimension of the layer and $h_{\text{out}}$ is the output dimension.

- **Edge-wise operations**: These involve aggregating the features of neighboring nodes through a message-passing operation over edges. This has a time complexity of $\mathcal{O}(e \cdot h_{\text{out}})$.

Step 2: Time Complexity of Each GCN Layer

For the $i$-th GCN layer:

- Let the input feature dimension be $h_{i-1}$ and the output feature dimension be $h_i$.
- Node-wise multiplication has complexity $\mathcal{O}(n \cdot h_{i-1} \cdot h_i)$.
- Edge-wise aggregation has complexity $\mathcal{O}(e \cdot h_i)$.

Thus, the total time complexity of the $i$-th layer is:

$$\mathcal{O}(n \cdot h_{i-1} \cdot h_i + e \cdot h_i)$$

Step 3: Summing Over All GCN Layers

We have $k$ GCN layers with dimensions $h_0, h_1, \ldots, h_k$, where $h_0 = n$ is the input feature dimension and $h_k = d$ is the output dimension. Therefore, the total time complexity for all layers is:

$$T_{\text{GCN}} = \sum_{i=1}^{k} \left( \mathcal{O}(n \cdot h_{i-1} \cdot h_i + e \cdot h_i) \right)$$

Step 4: VGAE Encoder and Decoder

- **Encoder**: The encoder, which maps node features to a latent representation space (mean and variance for the latent variables), has the same complexity as the GCN layers, so its complexity is $T_{\text{GCN}}$.

- **Decoder**: In VGAE, the decoder typically involves reconstructing the adjacency matrix from the latent space. The reconstruction (e.g., using a dot product between latent vectors) has a time complexity of $\mathcal{O}(n^2 \cdot d)$, as it involves calculating pairwise similarities between all node pairs.

Step 5: Total Time Complexity of VGAE

Summing up the time complexity of the GCN-based encoder and the decoder, we get the overall time complexity:

$$T_{\text{VGAE}} = T_{\text{GCN}} + \mathcal{O}(n^2 \cdot d)$$

This expands to:

$$T_{\text{VGAE}} = \sum_{i=1}^{k} \left( \mathcal{O}(n \cdot h_{i-1} \cdot h_i + e \cdot h_i) \right) + \mathcal{O}(n^2 \cdot d)$$

Conclusion

Let us denote $\max_{i=1}^{k} h_i$ by $h_{\max}$. We know that $n = h_0 \geq h_1 \geq \ldots \geq h_k = d$. So, $h_{\max} = h_1$ and the time complexity of the VGAE is:

$$T_{\text{VGAE}} = \mathcal{O}\left( \sum_{i=1}^{k} (n \cdot h_{i-1} \cdot h_i + e \cdot h_i) + n^2 \cdot d \right) \in \mathcal{O}\left( n^2 \cdot h_{\max} \right)$$

$$s.t. \; h_0 = n, h_k = d$$

This reflects the complexities of both the encoder (GCN layers) and the decoder (adjacency matrix reconstruction). The most significant term depends on the number of nodes, and the dimensions of the latent space. Hyperparameters of the VGAE model and the values assigned to them in the original paper are listed in Table S2.

| Method | Hyperparameter | Description / Value |
|--------|----------------|---------------------|
| VGAE | Latent Dimension ($d$) | Size of the latent space (dimension of node embeddings) (8) |
| | Graph Convolutional Layers (GCN) | Number of convolution layers to capture graph structure (2 layers) |
| | Learning Rate | Learning rate for the Adam optimizer (1e-2) |
| | Hidden Dimension ($h$) | Number of hidden units in the encoder GCN layers (32) |

Table S3: Hyperparameters for Variational Graph Autoencoder (VGAE)

### A.10.3 DYNGEM:

DynGEM uses a Multi-Layer Perceptron (MLP) autoencoder to generate low-dimensional embeddings for dynamic graphs at each snapshot. At time step $t = 1$, the model is trained on the first snapshot of the graph using a randomly initialized deep autoencoder. For subsequent time steps, embeddings and network parameters are initialized from the previous time step.

Given $n$ nodes, $k$ hidden layers with sizes $h_1, h_2, \ldots, h_k$, and the latent representation dimension $d$, the time complexity of processing the input graph for each snapshot is:

$$\mathcal{O}(n \cdot (n \cdot h_1 + h_1 \cdot h_2 + \cdots + h_{k-1} \cdot h_k + h_k \cdot d))$$

Conclusion

Let us denote $\max_{i=1}^{k+1} h_i$ by $h_{\max}$. We know that $n = h_0 \geq h_1 \geq \ldots \geq h_{k+1} = d$. So, $h_{\max} = h_1$ and the time complexity of the DynGEM is:

$$T_{\text{DynGem}} = \mathcal{O}\left(\sum_{i=1}^{k+1}(n \cdot h_{i-1} \cdot h_i)\right) \in \mathcal{O}\big(n^2 \cdot h_{\max}\big)$$

$$s.t.\ h_0 = n, h_{k+1} = d$$

Hyperparameters of this method and the assigned values to them can be found in Table S4.

| Method | Hyperparameter | Description / Value |
|--------|----------------|---------------------|
| DynGEM | Latent Dimension ($d$) | Size of the latent space (dimension of node embeddings) (8) |
| | Number of layers in the encoder/decoder | Autoencoder has 3 layers |
| | Layer Sizes ($h_1, h_2$) | Size of each layer in the autoencoder (500,300) |
| | L1 regularization coefficient ($\nu_1$) | Encourages sparsity in the model's weights ($1e-6$) |
| | L2 regularization coefficient ($\nu_2$) | Encouraging weight values to remain small ($1e-6$) |
| | Learning Rate | Learning rate ($1e-4$) |
| | Reconstruction Loss Weight ($\beta$) | Weight for adjacency matrix reconstruction (5) |

Table S4: Hyperparameters for DynGEM

### A.10.4 DYNAE:

DynAE extends a static MLP autoencoder to handle temporal graphs. It uses $l$ look-back adjacency matrices from past snapshots and feeds them into a deep autoencoder to reconstruct the current graph based on previous graphs.

Given an input size of $n \cdot l$ (where $n$ is the number of nodes and $l$ is the number of leook-back snapshots), and $k$ layers in the autoencoder, with the latent representation dimension $d$, the time complexity for the encoder is:

$$\mathcal{O}(n \cdot (n \cdot l \cdot h_1 + h_1 \cdot h_2 + \cdots + h_k \cdot d)$$

Conclusion

Let us denote $\max_{i=1}^{k+1} h_i$ by $h_{\max}$. We know that $n.l = h_0 \geq h_1 \geq \ldots \geq h_{k+1} = d$. So, $h_{\max} = h_1$. In addition, $l$ can be considered as a constant number, and the time complexity of the DynAE is:

$$T_{\text{DynAE}} = \mathcal{O}\left(\sum_{i=1}^{k+1}(n \cdot h_{i-1} \cdot h_i)\right) \in \mathcal{O}(n^2 \cdot h_{\max})$$

$$s.t.\ h_0 = n \cdot l, h_{k+1} = d$$

Hyperparameters of this method and the assigned values to them can be found in Table S5.

| Method | Hyperparameter | Description / Value |
|---|---|---|
| DynAE | Look-back ($l$) | Number of previous snapshots used (2) |
| | Latent Dimension ($d$) | Size of the latent space (dimension of node embeddings) (8) |
| | Number of layers in the encoder/decoder | Autoencoder has 3 layers |
| | Layer Sizes ($h_1, h_2$) | Size of each autoencoder layer (500,300) |
| | L1 regularization coefficient ($\nu_1$) | Encourages sparsity in the model's weights ($1e - 6$) |
| | L2 regularization coefficient ($\nu_2$) | Encouraging weight values to remain small ($1e - 6$) |
| | Learning Rate | Learning rate ($1e - 4$) |
| | Reconstruction Loss Weight ($\beta$) | Weight for adjacency matrix reconstruction (5) |

Table S5: Hyperparameters for DynAE

### A.10.5 DYNRNN:

DynRNN is similar to DynAE, but it uses Recurrent Neural Networks (RNNs), specifically Long Short-Term Memory (LSTM) networks, to capture temporal dependencies across snapshots. Each node's neighborhood at each snapshot is passed into the LSTM.

The time complexity for LSTM step $i$ on one node is:

$$\mathcal{O}(h_{i-1_{LSTM}} \cdot h_{i_{LSTM}} + h_{i_{LSTM}}^2)$$

Given $n$ nodes, $k_{LSTM}$ LSTM layers with sizes $h_{1_{LSTM}}, h_{2_{LSTM}}, \ldots, h_{k_{LSTM}}$ and $l$ snapshots, the total time complexity for one snapshot is:

$$\mathcal{O}\big(n \cdot (n \cdot l \cdot h_{1_{LSTM}} + h_{1_{LSTM}} \cdot h_{2_{LSTM}} + \cdots + h_{k-1_{LSTM}} \cdot h_{k_{LSTM}} + h_{k_{LSTM}} \cdot d + h_{1_{LSTM}}^2 + \cdots + h_{k_{LSTM}}^2 + d^2)\big)$$

Conclusion

Let us denote $\max_{i=1}^{k+1} h_{i_{LSTM}}$ by $h_{\max}$. We know that $n \cdot l = h_{0_{LSTM}} \geq h_{1_{LSTM}} \geq \ldots \geq h_{k+1_{LSTM}} = d$. So, $h_{\max} = h_{1_{LSTM}}$. Is addition, $l$ can be considered as a constant number, the time complexity of the DynRNN is:

$$T_{\text{DynRNN}} = \mathcal{O}\left(\sum_{i=1}^{k+1}(n \cdot (h_{i-1_{LSTM}} \cdot h_{i_{LSTM}} + h_{i_{LSTM}}^2))\right) \in \mathcal{O}(n^2 \cdot h_{\max})$$

$$s.t.\ h_{0_{LSTM}} = n \cdot l, h_{k+1_{LSTM}} = d$$

Hyperparameters of this method and the assigned values to them can be found in Table S6.

| Method | Hyperparameter | Description / Value |
|---|---|---|
| DynRNN | Look-back ($l$) | Number of previous snapshots used (2) |
| | Latent Dimension ($d$) | Size of the latent space (dimension of node embeddings) (8) |
| | Number of RNN Layers | Number of stacked LSTM layers (3) |
| | Hidden State Size | Number of hidden units in LSTM (500,300) |
| | L1 regularization coefficient ($\nu_1$) | Encourages sparsity in the model's weights ($1e-6$) |
| | L2 regularization coefficient ($\nu_2$) | Encouraging weight values to remain small ($1e-6$) |
| | Learning Rate | Learning rate ($1e-4$) |
| | Reconstruction Loss Weight ($\beta$) | Weight for adjacency matrix reconstruction (5) |

Table S6: Hyperparameters for DynRNN

### A.10.6 DYNAERNN:

DynAERNN combines the autoencoder from DynAE with the LSTM-based RNN from DynRNN. The encoder compresses the neighborhood vectors of $l$ snapshots into a low-dimensional space, which the LSTM processes across time to capture temporal dependencies.

The total time complexity for DynAERNN is the sum of the autoencoder and LSTM complexities:

$$\mathcal{O}(n \cdot (n \cdot l \cdot h_1 + h_1 \cdot h_2 + \cdots + h_{k-1} \cdot h_k)+$$

$$\mathcal{O}\big(n \cdot (h_k \cdot h_{1_{LSTM}} + h_{1_{LSTM}} \cdot h_{2_{LSTM}} + \cdots + h_{k-1_{LSTM}} \cdot h_{k_{LSTM}} + h_{k_{LSTM}} \cdot d + h_{1_{LSTM}}^2 + \cdots + h_{k_{LSTM}}^2 + d^2)\big)$$

Conclusion

Let us denote $\max(\max_{i=1}^{k} h_i, \max_{i=1}^{k+1} h_{i_{LSTM}})$ by $h_{\max}$. We know that $n \cdot l = h_0 \geq h_1 \geq \ldots \geq h_k = h_{0_{LSTM}} \geq h_{1_{LSTM}} \geq \ldots \geq h_{k+1_{LSTM}} = d$. So, $h_{\max} = h_1$. In addition, $l$ can be considered as a constant number time complexity of the DynRNN is:

$$T_{\text{DynAERNN}} = \mathcal{O}\left(\sum_{i=1}^{k}(n \cdot h_{i-1} \cdot h_i) + \sum_{i=1}^{k+1}(n \cdot (h_{i-1_{LSTM}} \cdot h_{i_{LSTM}} + h_{i_{LSTM}}^2))\right) \in \mathcal{O}\big(n^2 \cdot h\big)$$

$$s.t. \ h_0 = n \cdot l, h_{0_{LSTM}} = h_k, h_{k+1_{LSTM}} = d$$

Hyperparameters of this method and the assigned values to them can be found in Table S7.

### A.10.7 TAVRNN:

The time complexity of the TAVRNN framework is driven by several components, including GNN layers, GRU operations, and an attention mechanism. Below, we break down the total complexity into the time complexity of each component.

1. GNN and GRU Layers:

At each time step $t$, the model processes the graph using a combination of GNN layers and a GRU-based RNN. The time complexity for these operations can be broken down as follows:

- **Low-dimensional Embedding**: first of all, each $n$-dimensional neighborhood vector is mapped to a $h_{GRU}$-dimensional embedding using a one layer feed forward network. The time complexity of this part will be:

$$\mathcal{O}(n^2 \cdot h_{GPU})$$

| Method | Hyperparameter | Description / Value |
|---|---|---|
| DynAERNN | Look-back ($l$) | Number of previous snapshots used (2) |
| | Latent Dimension ($d$) | Size of the latent space (dimension of node embeddings) (8) |
| | Autoencoder Layer Sizes | Size of each autoencoder layer (500,300) |
| | Number of RNN Layers | Number of stacked LSTM layers (3) |
| | LSTM Hidden State Size | Number of hidden units in LSTM (500,300) |
| | L1 regularization coefficient ($\nu_1$) | Encourages sparsity in the model's weights ($1e-6$) |
| | L2 regularization coefficient ($\nu_2$) | Encouraging weight values to remain small ($1e-6$) |
| | Learning Rate | Learning rate ($1e-4$) |
| | Reconstruction Loss Weight ($\beta$) | Weight for adjacency matrix reconstruction (5) |

Table S7: Hyperparameters for DynAERNN

- **Graph Convolution (GNN)**: Similar to the VGAE mentioned above , the time complexity of the GNN layer is:

$$T_{\text{GNN}} = \sum_{i=1}^{k} \left( \mathcal{O}(n \cdot h_{i-1} \cdot h_i + e \cdot h_i) \right)$$

- **GRU Operation**: Since the inner functions of our GPU cell is implemented by GCN layers, the dominant term in the time complexity of the GPU cell in each time step is equal to:

$$\mathcal{O}(n \cdot h_{GRU}^2 + e \cdot h_{GRU})$$

2. Temporal Attention Mechanism:

The attention mechanism aggregates past hidden states over a window of size $w$. The attention of the model into the last $w$ snapshots is computed in:

$$\mathcal{O}(w \cdot h)$$

where $w$ is the attention window size and $h$ is the hidden dimension. The time complexity of computing the weighted average vectors for all the $n$ node according to these computed attentions is:

$$\mathcal{O}(n \cdot w \cdot h)$$

3. Reconstruction: Similar to VGAE, the reconstruction process in TAVRNN is through computing the inner product of the final representation of each pair of the nodes, and its time complexity is:

$$\mathcal{O}(n^2 \cdot d)$$

4. Overall Time Complexity for Each Time Step:

The overall time complexity at each time step is a combination of the initial projection to a low-dimensional space using a feedforward layer, GNN and GRU computations, attention mechanism, and reconstruction:

$$\mathcal{O}(n \cdot (h_1 + h_1 \cdot h_2 + \cdots + h_k \cdot d) + e \cdot (h_1 + \cdots + h_k) + n \cdot h_{GRU}^2 + e \cdot h_{GRU} + (n+1) \cdot w \cdot h + n^2 \cdot d)$$

Conclusion

Let us denote $\max(\max_{i=1}^{k+1} h_i, h_{GRU}, h)$ by $h_{\max}$. We know that $n \cdot l = h_0 \geq h_1 \geq \ldots \geq h_k + 1 = d$. So, $h_{\max} = h_1$. We can infer that the time complexity of TAVRNN is:

$$T_{\text{TAVRNN}} = \mathcal{O}\left(\sum_{i=1}^{k+1}(n \cdot h_{i-1} \cdot h_i + e \cdot h_i) + n \cdot h_{GRU}^2 + e \cdot h_{GRU} + n \cdot w \cdot h + n^2 \cdot d\right) \in \mathcal{O}\left(n^2 \cdot h_{\max} + n \cdot w \cdot h\right)$$

$$s.t.\ h_0 = 1, h_{k+1} = d$$

The summary of the time complexities for different methods is shown in Table S8.

Table S8: One forward pass time complexity for one time window (i.e. snapshot).

| Method | Complexity |
|---|---|
| VGAE | $\mathcal{O}\left(\sum_{i=1}^{k}(n \cdot h_{i-1} \cdot h_i + e \cdot h_i) + n^2 \cdot d\right) \in \mathcal{O}\left(n^2 \cdot h_{\max}\right)$ |
| DynGEM | $\mathcal{O}\left(\sum_{i=1}^{k+1}(n \cdot h_{i-1} \cdot h_i)\right) \in \mathcal{O}\left(n^2 \cdot h_{\max}\right)$ |
| DynAE | $\mathcal{O}\left(\sum_{i=1}^{k+1}(n \cdot h_{i-1} \cdot h_i)\right) \in \mathcal{O}\left(n^2 \cdot h_{\max}\right)$ |
| DynRNN | $\mathcal{O}\left(\sum_{i=1}^{k+1}(n \cdot (h_{i-1_{\text{LSTM}}} \cdot h_{i_{\text{LSTM}}} + h_{i_{\text{LSTM}}}^2))\right) \in \mathcal{O}\left(n^2 \cdot h_{\max}\right)$ |
| DynAERNN | $\mathcal{O}\left(\sum_{i=1}^{k}(n \cdot h_{i-1} \cdot h_i) + \sum_{i=1}^{k+1}(n \cdot (h_{i-1_{\text{LSTM}}} \cdot h_{i_{\text{LSTM}}} + h_{i_{\text{LSTM}}}^2))\right) \in \mathcal{O}\left(n^2 \cdot h_{\max}\right)$ |
| GraphERT | $\mathcal{O}((\gamma \cdot |p| \cdot |q| \cdot H \cdot k) \cdot n \cdot L^2 \cdot h_{\max}) \in \mathcal{O}\left(n \cdot L^2 \cdot h_{\max}\right)$ |
| TAVRNN | $\mathcal{O}\left(\sum_{i=1}^{k+1}(n \cdot h_{i-1} \cdot h_i + e \cdot h_i) + n \cdot h_{\text{GRU}}^2 + e \cdot h_{\text{GRU}} + n \cdot w \cdot h + n^2 \cdot d\right) \in \mathcal{O}\left(n^2 \cdot h_{\max} + n \cdot w \cdot h_{\max}\right)$ |

