# OpenReview forum: "TAVRNN: Temporal Attention-enhanced Variational Graph RNN Captures Neuronal Dynamics and Behavior"
_ICLR.cc/2025/Conference — ICLR 2025 Conference Withdrawn Submission_

### Official Review · Reviewer_ns94 · 2024-10-27

**Soundness:** 1
**Presentation:** 1
**Contribution:** 1
**Rating:** 1
**Confidence:** 4

**Summary:**

The authors present a graph RNN-based method for extracting insights from time-varying functional connectivity networks. They apply their approach to recordings of neural activity from two real-world datasets, and demonstrate good performance in classification and clustering tasks. Finally, they underscore the computational efficiency of their method.

**Strengths:**

The authors demonstrate that their proposed method performs comparably or better to other methods in classifying or clustering relevant behavioral states from observed recorded activity.

**Weaknesses:**

Much of the methods section appears to have been directly lifted from "A Deep Probabilistic Spatiotemporal Framework for Dynamic Graph Representation Learning with Application to Brain Disorder Identifcation", Yap et al., IJCAI 2024, with no reference to this work.

Beyond that, the paper is poorly organized. The proposed method, as well as its evaluations, are described in an inscrutable manner. What exactly is being modeled? (Why are we extracting latent features from binarized functional connectivity matrices?) What exactly is used as node features? How are predictions (e.g. for classification or clustering) read out from the model?

**Questions:**

See above.

**Details Of Ethics Concerns:**

Much of the methods section has been directly lifted from "A Deep Probabilistic Spatiotemporal Framework for Dynamic Graph Representation Learning with Application to Brain Disorder Identifcation", Yap et al., IJCAI 2024, with no reference to this work.

---

> ### Author Response · Authors · 2024-11-16
>
> - We acknowledge that referencing Yap et al., IJCAI 2024, in our manuscript would be appropriate, and we will ensure that it is added to our list of references. However, we feel that raising an ethics flag and making accusations of plagiarism is unfair and unwarranted.
> Our work indeed builds upon established methodologies, as is common in scientific research, to advance the field. We have clearly cited the foundational work by Hajiramezanali, Ehsan, et al., "Variational Graph Recurrent Neural Networks," which is a more seminal and widely recognized study that forms the basis for our proposed method. This same work is also cited in the Yap et al. paper, which itself extends upon these earlier concepts.
> While we appreciate that Yap et al. have made additional contributions, we emphasize that our proposed temporal attention mechanism is conceptually and structurally distinct. Specifically, our attention layer considers entire temporal graphs and captures historical graph information, unlike Yap et al.'s node-based attention at individual time steps. We respectfully maintain that there is no basis for ethical concerns, as our work represents a genuine and novel extension of existing methods in line with the norms of scientific advancement.
>
> - Thank you for your feedback. We appreciate the opportunity to clarify the focus and organization of our work. TAVRNN is designed to model temporal changes in network structure and extract latent representations for each individual unit (e.g., recording electrodes in the DishBrain dataset or neurons in the calcium imaging data) at every time point. These latent representations allow us to track how cluster formations and node associations evolve over time, providing insights into the dynamic mechanisms underlying behavioral performance.
> The binarized functional connectivity matrices serve as adjacency matrices ($A_t$​), indicating the existence of connections between nodes, while the true values of the functional connectivity (i.e., connection weights) are incorporated as node features ($X_t$​). This dual-input approach ensures that the model captures both the structural connectivity (presence of links) and the strengths of these connections. This is critical for retaining the nuances of network dynamics while simplifying the computational complexity of modeling the overall network.
> Our decision to use binarized adjacency matrices reflects the goal of identifying high-level patterns in network structure while focusing the model's capacity on learning the dynamics encoded in the connection strengths. By incorporating the true connectivity weights as node features, TAVRNN ensures that all meaningful information about the functional network is preserved for downstream analysis. These node features, derived from the functional connectivity inference method specific to the dataset (as detailed in Section 4.1), include the weights of existing connections for each node, enabling the model to leverage both structural and functional aspects of the data.
> We also acknowledge that Figure S4, which effectively illustrates how TAVRNN captures temporal changes corresponding to different behavioral performance levels (e.g., gameplay vs. rest), deserves greater prominence. Following suggestions from other reviewers, we will move this figure to the main text and expand the accompanying discussion to make the method and its evaluations more accessible. These revisions, along with a reorganization of the manuscript for improved clarity, will better articulate what is being modeled and how the node features contribute to the analysis.
>
> - Thank you for your question. Predictions for classification or clustering tasks are based on the temporal node embeddings ($Z_t$​), which represent lower-dimensional representations extracted from the recorded neural activity at each time step. These embeddings are generated by leveraging information from $A_t$​ (adjacency matrices), $X_t$​ (node features), and $H_{t-1}$​ (history matrix), providing a rich representation of the dynamic relationships between nodes.
> The embeddings ($Z_t$​) were evaluated through various analyses detailed in the manuscript, including classification, clustering, and qualitative assessments like t-SNE visualizations, to demonstrate their utility and interpretability. Although the model optimizes a reconstruction loss for $A_t$​ during training (see Eq. 4), the primary emphasis is on the quality of $Z_t$​ as a robust and meaningful representation of neuronal and behavioral dynamics.
> To improve clarity, we will add explicit descriptions of how predictions are derived from $Z_t$​ in the methods section and expand the caption of Figure 2 to highlight this process more clearly.

---

> > ### Comment · Reviewer_ns94 · 2024-11-25
> >
> > I appreciate the authors' responses. However, I unfortunately still remain unconvinced that the work is suitable for publication at a venue of the level of ICLR. While I appreciate some of the clarifications, I have yet to see an updated manuscript during this revision period that addresses my strong concerns regarding the clarity of presentation. That aside, I share concerns raised with other reviewers (PQUB12) over the minimal level of novelty and contribution, as well as concerns about the necessity of this particular complex modelling pipeline being poorly motivated (Vtt6). I also address some of the author's responses below:
> >
> > - The vast majority of the methods section of the present manuscript, down to the notation used, is essentially identical to that of Yap et al., IJCAI 2024. As such, I maintain that it is a glaring and misleading omission to not cite that work, as doing such obfuscates, if not misrepresents, the scope of contributions here. I do appreciate that the authors now agree to include a citation of that work. Nonetheless, the fact that details about the claimed novel contribution of the present work--namely, adding temporal attention over $H_t$--are relegated to the appendix, rather than included in the lengthy methods section of the main text, betrays the fact that the contribution added by this work appears rather limited.
> >
> > - I appreciate the clarification regarding how exactly features are readout for downstream tasks. Nonetheless, some things remain unclear. Are the features $Z_t$ used directly for performance quantification in clustering, or are those evaluations being computed on top of the further dimensionality-reduced t-SNE coordinates? Especially if it is the latter, why not simply evaluate classification performance of predicting the correct channel (Sens, R_{down}, etc.) from a linear readout of these features?

---

> > > ### Author Response · Authors · 2024-12-03
> > >
> > > 1. We agree that the fundamentals of TAVRNN are built upon foundational works such as Spatiotemporal Variational Bayes, Recurrent Neural Networks, and etc., as also presented in Yap et al. and other literature in the field. Once again, our current work builds upon established methodologies, a standard and highly common practice in our field. We have appropriately cited foundational works, including "Variational Graph Recurrent Neural Networks" by Hajiramezanali et al., which is a key reference for both our current submission and the study by Yap et al. While we are willing to include Yap et al. in our references for added clarity, it is essential to emphasize that our proposed method is distinct. This work introduces a novel temporal attention mechanism that captures full temporal graphs and historical dependencies, in contrast to the node-based attention approach in Yap et al. This level of contribution, which involves building on prior foundational methods while introducing a novel mechanism, is highly standard and widely accepted within our research community.
> > > These fundamentals are discussed in identical terms across the literature since they represent standard concepts and cannot be altered. Moreover, while acknowledging that omitting this paper from the reference list was an oversight during the revision process and we appreciate the reminder to complete our reference list, importantly, given that there is actually an overlap in the author lists between these two manuscripts, we find this accusation of (self-)plagiarism not only unjustified but also deeply unfair. Furthermore, as Variational Graph RNNs are widely applied in this field in various contexts, specifically pinpointing only our previous work (Yap et al.) in this double-blind peer-review process is concerning to us.
> > > As appreciated by the reviewer, the methodological novelty of this work lies in the design of the temporal attention mechanism. While this is currently detailed in the appendix, we are happy to move this discussion to the main text in future iterations to further emphasize the contribution.
> > >
> > >
> > > 2. To clarify, the embedding features (Z_t ) are directly used for performance quantification. The t-SNE visualization is solely employed as a dimensionality reduction tool for 2D visualization, aiming to effectively communicate the findings in a more engaging and accessible manner. It is not used for performance evaluation or classification tasks.

---

### Official Review · Reviewer_yW3e · 2024-11-03

**Soundness:** 3
**Presentation:** 2
**Contribution:** 2
**Rating:** 5
**Confidence:** 4

**Summary:**

This work proposed a nonlinear and stochastic graph unsupervised representation learning to study neural activities data. This model integrates an attention-enhanced graph neural network to construct node representation from dynamical neuronal connectivity, temporal components are extracted using recurrent neural network and temporal attention mechanism to learn a time-dependent state matrix. It further incorporates variational inference frameworks with probabilistic node embedding to reflect uncertainty. The learned representation from this work is evaluated on two datasets including in-vivo calcium imaging from rate to decode position, and in-vitro ephys data from Dish Brain to study structure changes during adaptive learning.

**Strengths:**

1. This work focuses on a novel and important prospective to incorporate a dynamic view of neuron interaction to learn a temporal adapted graph embeddings with unsupervised learning, which allows study temporal structure changes during different behaviors.
2. It introduces a novel dataset and benchmark from DishBrain to study how self-organization structure emerge during learning.
3. The work is compared against multiple extensive graph representation baselines, and outperforms others the clustering patterns in the DishBrain datasets, and demonstrate it is more scalable with advantages in time complexity.
4. Comprehensive ablation studies are performed.

**Weaknesses:**

1. TAVRNN's predictive performance is limited compared to GraphERT in classifying rat's position, as shown in Table 1.
2. The evaluation and benchmarking with other baselines are only based on graph representation approaches, which takes the functional connectivity with measured with zero-lag pearson correlation to start. However, the linearity assumed in this correlation metrics often ignore high-order statistics and nonlinear information in neural dynamics. Others dynamical modeling approaches such as LFADS or self-supervised contrastive learning approaches like CEBRA which directly take the time series as inputs and implemented in a more end-to-end fashion could also be potentially applied to the same tasks. The benefits of learning graph representation are not evaluated and remain unclear.
3. It is not clear in the section 5, how does the downstream task evaluations are performed, whether the downstream tasks only evaluate $Z_t$ or memory embedding $H_t$ as well? How are the train/val/test data split designed, whether cross-session, single- or multi-animal are evaluated, more implementation details should be included.
4. Presentation and writing needs improvement.

[1] Inferring single-trial neural population dynamics using sequential auto-encoders. 2018

[2] Learnable latent embeddings for joint behavioural and neural analysis. 2023

**Questions:**

1. Add more clarification on what does $H_t$ represent? How it will be evaluated?

---

> ### Author Response · Authors · 2024-11-16
> **Author responses to reviewer yW3e comments (part1)**
>
> We would like to thank the reviewer for their evaluation of the manuscript and comments and suggestions.
>
> - Thank you for your feedback. We acknowledge that TAVRNN's predictive performance on the rat position classification task is slightly lower than that of GraphERT, as shown in Table 1. However, the primary purpose of including this dataset was validation. The calcium imaging dataset provides observable behavior and clear ground truth labels (the animal's position), making it an ideal controlled environment for rigorously testing the model’s foundational capabilities before applying it to more complex datasets.
> Our main objective was to develop a framework capable of capturing nuanced associations between neural activity and behavior in challenging scenarios like the DishBrain dataset, where ground truth labels are unavailable, and the mechanisms underlying performance levels are not fully understood. In these complex settings, TAVRNN consistently outperforms all baselines, including GraphERT, and provides additional insights by uncovering clusters of electrodes associated with enhanced behavioral performance. These insights into network structure and dynamics are not captured by GraphERT or other baselines, showcasing TAVRNN's utility beyond predictive accuracy.
> Additionally, TAVRNN is significantly more efficient in terms of computational complexity compared to GraphERT, as demonstrated in our time complexity analysis. This efficiency is particularly valuable when scaling to large, high-dimensional datasets like DishBrain. While GraphERT demonstrates strong classification performance in simpler tasks, TAVRNN's broader applicability, interpretability, and computational efficiency highlight its distinct advantages in addressing more complex scientific questions. We will clarify these points further in the revised manuscript to provide a more comprehensive comparison.
>
> - Thank you for your feedback. As described in Section 4.1 of the main text, with additional details in Appendix A.5.1, we carefully selected the connectivity inference method for the DishBrain dataset using graph kernels to ensure the most effective representation of the network structure. We appreciate your mention of other approaches like LFADS and CEBRA, and we will ensure that LFADS is discussed in our literature review, alongside the already-cited CEBRA.
> It is important to note that these methods have distinct goals compared to ours. Methods like LFADS and CEBRA excel at capturing population-level neuronal trajectories using full temporal spiking data in an end-to-end manner. However, these approaches are not designed to preserve information about individual units (neurons or electrodes) or the dynamic associations between nodes over time. Our study specifically targets this gap, focusing on understanding how changes in node-level associations relate to variations in behavioral performance.
> In scenarios like the DishBrain dataset, where we already observe behavioral metrics such as the hit-to-miss ratio indicating population-level changes, our aim is to delve deeper into the mechanisms driving these changes. While methods like CEBRA effectively capture shifts in population trajectories, they do not explain the underlying node-level dynamics contributing to performance differences. TAVRNN, with its graph-based representation, enables the exploration of these finer-grained dynamics, including the evolving roles of individual nodes and their associations, which is critical for understanding task-specific neuronal adaptations.
> Furthermore, the graph-based approach offers a unique advantage by modelling functional connectivity as a dynamic network, allowing us to uncover patterns of synchronization, modularity, and clustering that align with behavioral variations. These insights go beyond population-level trajectory detection by providing interpretable representations of the neural mechanisms driving performance. Such granularity is essential for answering specific research questions about the role of network structure in behavior, which methods like LFADS and CEBRA are not designed to address.
> While we recognize the strengths of LFADS and CEBRA in analyzing population-level dynamics, their scope differs fundamentally from our focus on graph-based representations. We will incorporate this discussion into the revised manuscript to clarify the complementary yet distinct benefits of TAVRNN for investigating dynamic neural networks and their relationship to behavior.

---

> > ### Author Response · Authors · 2024-11-16
> > **Continuation of author responses to reviewer yW3e comments (part2)**
> >
> > - Thank you for your comment and for highlighting these important points. These embeddings were specifically designed to capture dynamic and structural properties of individual nodes in the network, making them the focus of our evaluations. While the memory embeddings $H_t$​ play a critical role in capturing historical information and enriching $Z_t$​ during training, the downstream tasks only evaluate $Z_t$​ as the primary output. We will ensure this distinction is explicitly clarified in the main text.
> > Regarding the experimental setup, we have detailed aspects of the implementation in Section A.9 and related sections of the supplemental material. However, we acknowledge the need to make these details more accessible in the main text. To address your concerns, we will include a clearer explanation of how the train/validation/test splits were designed. Specifically, our evaluations were conducted using single-animal or single-dish recordings without cross-session or cross-animal validations. This design allowed us to rigorously assess the temporal dynamics within a controlled context. We recognize the importance of exploring cross-session or multi-animal generalization as a potential future direction and will discuss this in the revised manuscript.
> >
> > - Thank you for your feedback on the presentation and writing. We recognize the importance of clear and concise communication, and we will take this opportunity to improve the overall clarity and readability of the manuscript. In addition to addressing the specific points raised in response to previous comments, we will carefully revise the text to ensure that the explanations, structure, and flow are more accessible to readers.
> > We will also refine the presentation of figures, tables, and supplementary materials to ensure they complement the narrative effectively. These changes, combined with the inclusion of additional clarifications and implementation details, aim to enhance the overall quality of the manuscript. We appreciate your constructive feedback and will make these improvements in the revised version.
> >
> > - Thank you for pointing out the requirement for more clarification about $H_t$. We appreciate the opportunity to clarify. $H_t$ represents the history matrix, which encapsulates the accumulated temporal information from previous time steps. It serves as a memory embedding that allows the model to capture the evolving dynamics of neuronal activity over time. $H_t$ plays a crucial role in enriching the latent embeddings $Z_t​$ by incorporating temporal dependencies, enabling the model to represent both the current and historical states of the system.
> > While $H_t$​ itself is not directly evaluated, its contribution is assessed indirectly through the evaluation of the learned latent embeddings $Z_t​$. These embeddings, constructed using $H_t$​ alongside $A_t​ $ and $X_t​$, are designed to capture the underlying neural dynamics and their relationships with behavioral outcomes. The utility and interpretability of $Z_t$​ are evaluated through various analyses, including classification, clustering, and qualitative assessments (e.g., t-SNE visualizations), as presented in the manuscript.
> > To further clarify this in the revised manuscript, we will explicitly describe the role of $H_t​$ in the embedding process and emphasize its contribution to the effectiveness of $Z_t​$. We appreciate your suggestion and will ensure these details are clearly articulated.

---

> > > ### Comment · Reviewer_yW3e · 2024-11-25
> > >
> > > Thank you to the authors for their responses and clarifications. The focus on node-level representation instead of population-level representation is reasonable and makes sense to me. However, I believe the presentation still requires significant improvement, and the evaluations are limited to a single-session dataset. Additionally, the novelty appears incremental, and the improvements over existing works seem modest. Thus I have maintained my current score.

---

### Official Review · Reviewer_Vtt6 · 2024-11-04

**Soundness:** 2
**Presentation:** 2
**Contribution:** 2
**Rating:** 3
**Confidence:** 3

**Summary:**

This paper introduces a deep probabilistic latent dynamics model that uses graph RNNs and an attention mechanism to model the time evolution of pairwise correlations across a neural population in a lower dimensional embedding space. The approach is applied to a hippocampal rat data set, as well as a novel dataset involving in vitro brain culture trained by a closed-loop control MEA system to play a modified version of the pong video game. Ablation results are provided on variants of the model applied to the rat dataset. The paper shows insights into the brain culture population structure associated with varying performance in the game.

**Strengths:**

The paper combines a number of cutting-edge approaches (variational GRNNs, attention) and applies them to an exciting dataset involving in vitro closed-loop learning of a complex task, with potentially interesting insights.

**Weaknesses:**

The paper seems to be addressing an exciting application area with potential for very interesting insights into learning. However, given the serious limitations of this work I don’t recommend publishing this paper at ICLR in its present form.

I found the premise compelling and spent a lot of time trying to understand the paper, but found the paper presentation sprawling and unfocused, with the conceptual goals and key mathematical setup quite hard to follow. The conceptual figure explaining the model is fairly confusing as well. Claims on the performance and potential impact of the model in the paper seem substantial, and it seems unclear that they are well justified by the evaluations presented here.

If I followed correctly, it seems like a simple conceptual summary of the technical approach is: the authors want to model the time varying correlation structure of the network Xt at each timepoint, which seems like a good goal, by learning to reconstruct a thresholded adjacency matrix A. They would like to do this using a temporal latent variable model, ie by learning dynamics acting on a low dimensional latent space Z that summarizes this high dimensional structure. To facilitate this, they’ll also pass X, the unthresholded correlation matrices, to the latent dynamics function and inference network. An RNN will keep track of the history of the correlations going back beyond the previous timepoint. Confusingly, the Z dynamics function (transition network prior) operates on Ht or the summary of the past history provided by the inference RNN, so no explicit dynamics on Zt are learned (equation 8a). There is an attention mechanism applied to the history of H used to update Ht, further obfuscating the learned dynamics but potentially improving performance. This produces two latent state spaces of interest, one on Z and one on H, with line 295 referring to {Z,H} as the learned embedding spaces. It’s unclear how the two will be used for analysis: for example which of the two, or both, are used to visualize embeddings learned by the model.

It would be helpful to write down the (generative) graphical model involved, ie Zt -> Xt -> At, as well as the time evolution of z. If i understand correctly, At should be conditionally independent of Z when considering X, since the adjacency matrix is a thresholded version of the correlation features?

The model presented is quite complex, so it is important to explain the goals of the model, to motivate why the complexity of the proposed model is necessary to achieve these goals, and why the benchmarks used to evaluate the model are both appropriate for these goals and for showcasing the particular capabilities of the model.

**Goals of the model:**

- It’s unclear why learning the adjacency structure, vs the full, thresholded correlation matrix X is a desirable goal for the model, since we can always apply thresholding afterwards. Is this easier than learning the full correlations X? Is there another reason for framing this as the objective?

- Is the learned time dynamics function also of interest, and if so how is it made accessible / interpretable and used for analysis? If not, what are the advantages of learning a temporal model, vs fitting a static latent variable model like a graph VAE to each timepoint / snapshot separately? Performance on the classification metric on the rat dataset for the model is better than for VGAE, but VGAE also outperforms some of the other temporal models, suggesting this benchmark may not be a good test of the advantages of a temporal model (more on this below).

- It would be helpful to clarify what the desired capabilities of a learned embedding model are, and how these differ from the objective function used to train the network, as well as to motivate why it is advantagous not to include these downstream tasks in the objective in some way (eg generalization, emergent properties etc).

**Appropriate validation metrics:**

- The model is trained to reconstruct the thresholded connectivity matrices At, but no metrics on the ability of the model to reconstruct these using the inference approach seem to be provided. Additionally, no test of the generative quality of the learned dynamics, when run in forward or generative mode, is shown. It would be important to show how the model compares on these with other complex models for which results are presented, as well as simple baselines, such as static models like VAE/GVAE variants, or even PCA. It is possible that Table S1 in the supplement, which does not have a legend explaining it or a reference in the text that I could find, includes something like this kind of result?

- 5.1 - if I understood correctly, this benchmark is a binary classification task that involves decoding whether the data comes from the start or end of the track the mouse is on. It’s not clear why this is a desirable target benchmark from the perspective of showing this complex model's capabilities, or whether it is challenging or easy. A relevant simple baseline benchmark may be decoding these labels from the recordings themselves, from the raw correlation matrices computed on the raw data (eg Xt), or even from a projection onto the first d PCs of the raw data, where d is equivalent to the dimensionality used by the model. If this classification is easy to do from the raw data, then this test seems like a sanity check that the latents are encoding structure in the data, rather than a particular advantage of the model. It is also worth noting that another model, GraphERT, performs better in this benchmark on accuracy and other metrics except recall, where it is very close.

- 5.3 -  On this dataset as well it would be useful to show how the network performs on the task it was trained to do -- reconstructing the network adjacency matrices.

**Impact**

Supp. Figure S4 seems to be the most interesting and impactful scientific result, indicating that the model’s learned representation can give insight into network structure underlying variable performance, but this is hidden in the supplement.

**Questions:**

- What is the dimensionality typically used for Z (or H? if that is the important embedding space) and how is it chosen?
- figure 5 -- how are labeled channels being embedded in the latent space presented here? Wouldn’t each point in the embedding space represent an embedding of a full network structure At?

---

> ### Author Response · Authors · 2024-11-16
> **Author responses to reviewer Vtt6 comments (part1)**
>
> We would like to thank the reviewer for their evaluation of the manuscript and comments and suggestions.
>
> - The main readout of TAVRNN is ( $Z_t$ ), which represents the latent embedding of each individual recorded node (neurons or recording electrodes) and serves as the primary basis for further analysis. The primary goal of our work is to track these individual units over time to understand how associations and connections between nodes chenge and how these dynamics relate to behavioral performance. This allows us to gain deeper insights into the underlying mechanisms that influence performance levels. In Figure 5, each point in the latent space represents a single electrode (rather than the full network structure At), and the points are later color-coded based on their pre-assigned regions in the DishBrain system. We will ensure to clarify these details in the main text for greater conceptual clarity.
>
> - To write down the (generative) graphical model involved, we can write:
>
> ${A_t, X_t, H_{t-1}}$ → $(\mu_t^{enc},  \Sigma_t^{enc})$ → $Z_t$ (Equation 8b)
>
> ${Z_t, A_t, X_t, H_{t-1}}$ → $H_t$ (Equation 9)
>
> - Thank you for highlighting the importance of clearly articulating the goals of our model and justifying its complexity. We will ensure to provide a more detailed explanation of the motivations behind TAVRNN’s design and clarify how our chosen benchmarks appropriately showcase its unique capabilities.
> Briefly, the main impetus for developing the TVRNN framework stems from the limitations we observed in static or highly-dimensional connectivity analysis as shown in Figure 4). Simply examining connectivity patterns in networks — whether in the rat hippocampus neurons or  DishBrain system electrodes—did not adequately address the meaningful association between behavioral performance (such as Pong gameplay success in DishBrain) and the underlying neuronal dynamics. This approach alone lacked the temporal perspective needed to capture the evolving l relationships and network dynamics that likely drive performance changes.
> To bridge this gap, we designed TAVRNN to provide a more comprehensive view of these dynamic processes. The complexity of our model enables it to track and interpret the temporal changes in neuronal populations, offering a nuanced understanding of the relationships between network reorganization and behavioral outcomes. Our benchmarks were selected to validate these capabilities and to demonstrate TAVRNN’s effectiveness in uncovering these dynamics, which are crucial for understanding how neural network activity influences behavioral performance.
>
> - We appreciate the reviewer’s question and the opportunity to clarify our approach. Our method is not restricted to cases where matrix $A_t$ is derived directly from  $X_t$  via thresholding. Instead,   $X_t$  represents a general feature matrix, and the ground truth $A_t$  may arise from various sources, independent of a simple thresholding operation on $X_t$. In the context of our experiments, due to the inherent limitations of neuronal recordings, a direct ground truth  $A_t$ is not observable. This lack of direct reference matrix introduces additional challenges but also enriches the study’s scope. Consequently, we use thresholding on  $X_t$   as a heuristic to approximate $A_t$.
> Even in scenarios where  $X_t$  contains all the necessary information to compute  $A_t$, our objective of learning $A_t$ instead of $X_t$ offers computational and conceptual advantages. Learning $A_t​$ focuses on the critical structural relationships rather than the potentially noisy or redundant data in $X_t$​.  Furthermore, our experiments demonstrate that the latent representation $Z_t$, which encodes information from $A_t$, $X_t$, and $H_{t-1}$, are highly effective for downstream tasks such as clustering and classification. These representations encapsulate the temporal dynamics and interdependencies among nodes, providing insights that are both interpretable and task-relevant.
> We hope this addresses the concerns raised and grateful for the reviewer’s valuable feedback.

---

> > ### Author Response · Authors · 2024-11-16
> > **Continuation of author responses to reviewer Vtt6  comments (part2)**
> >
> > - We thank the reviewer for raising this point. The superior performance of TAVRNN compared to VGAE highlights the importance of learned temporal dynamics. Temporal modelling is crucial for understanding the behavior of neuronal populations, as these populations often carry biologically relevant information that static models cannot fully capture. For instance, _in vivo_ rat hippocampal neurons are well-known modes for studying memory and spatial navigation, while in vitro the neuronal cultures exhibit adaptive learning mechanisms over time. The added value of temporal modelling is further supported by our ablation study (Table 2), which highlights the significant performance gain attributed to the temporal attention mechanism.  These findings highlight that applying static latent variable models, which treat each snapshot independently, fails to fully capture the rich temporal interdependencies inherent in these datasets.
> > We also appreciate the reviewer’s observation regarding VGAE’s relative performance compared to some temporal models. While VGAE outperforms certain temporal methods on specific tasks like classification, this observation does not undermine the relevance of temporal models or the validity of our benchmark.  Different methods are designed for various objectives and their performance varies depending on datasets characteristics and task requirements. In fact, similar trends have been frequently reported in the literature (e.g., Beladev et al. "GraphERT--Transformers-based Temporal Dynamic Graph Embedding."; Hasanzadeh et al. and "Variational graph recurrent neural networks." Advances in neural information processing systems), where  VGAE occasionally outperforms temporal models in specific contexts. However, our results also show that certain temporal models, including TAVRNN, consistently achieve superior performance, particularly in clustering tasks, thereby demonstrating their utility. Overall, the observed variability in performance does not undermine the validity of our benchmark but rather emphasizes the nuanced strengths and limitations of different methods. By including a diverse set of models and tasks, our benchmark offers a comprehensive evaluation framework that captures the multifaceted nature of temporal and static approaches. We hope this addresses the reviewer’s concerns and thank them for their thoughtful feedback.
> >
> > - We appreciate the reviewer’s question regarding the desired capabilities of learned embedding models and their relationship to the training objective.   Broadly, embedding models are expected to achieve two main goals: 1) generate representations that are useful for downstream tasks, such as classification or clustering, and 2) provide interpretable embeddings that capture meaningful structural or temporal patterns in data. We address both goals in our study. For downstream tasks, we evaluate the embeddings through classification and clustering  (as shown in Table 1 and Table 4), demonstrating their utility in practical applications. For interpretability, we provide t-SNE visualizations (Figures 5, 6, and S4) to qualitatively illustrate how well the embeddings reflect inherent patterns in the data.
> > While it is possible to incorporate downstream task objectives into the model's learning process, this approach risks biasing the embeddings towards specific tasks, potentially at the expense of their generality.  One of the key strengths of unsupervised (or semi-supervised) models like ours is their ability to produce embeddings with emergent properties that generalize across diverse tasks. By decoupling the training objective from downstream tasks, we preserve the flexibility of the embeddings, ensuring they remain broadly applicable and not overfitted to a particular task. This separation also allows us to independently assess the quality of the embeddings, as their performance on downstream tasks serves as an unbiased evaluation metric.

---

> > > ### Author Response · Authors · 2024-11-16
> > > **Continuation of author responses to reviewer Vtt6  comments (part3)**
> > >
> > > - We appreciate the reviewer’s comment regarding the reconstruction of $A_t$ and the generative quality of learned dynamics. While TAVRNN optimizes a reconstruction loss for $A_t$ during training (see Eq. 4), the primary focus of our framework is not on faithfully reconstructing $A_t$​ but on leveraging the information in $A_t​, X_t​$, and $H_{t-1}​$ to produce meaningful latent embeddings ($Z_t$​). These embeddings are designed to capture the temporal and structural properties of the underlying neuronal activity, facilitating downstream tasks like clustering and classification. The interpretability and utility of $Z_t$ were central to our evaluation, as demonstrated in the main text through analyses such as classification (Tabel 1), clustering (Table 4), and quantitative t-SNE visualizations (Fig. 5, 6, and S4).
> > > Regarding the reconstruction quality of $A_t​$, while not a primary focus, we recognize that providing these metrics could strengthen the understanding of TAVRNN’s capabilities in comparison to other methods. We will include additional experiments in the revised manuscript that assess the reconstruction fidelity of $A_t​$ using our inference approach and benchmark it against static models like VAE variants and simpler baselines such as PCA. These analyses will complement the existing results by elucidating the model's ability to reconstruct and utilize $A_t​$ for learning dynamic embeddings.
> > > On the generative quality of the learned dynamics, we acknowledge the importance of testing the forward generative mode of TAVRNN. As a temporal model, TAVRNN's ability to simulate plausible future connectivity states or generate latent dynamics under varying conditions is a critical aspect of its utility. We will conduct additional experiments to test the generative performance and include comparisons with baseline models. These results will provide a more comprehensive assessment of TAVRNN's generative quality.
> > > Finally, Table S1, which the reviewer references, summarizes the classification effectiveness achieved through 5-fold cross-validation on the DishBrain dataset. The table reports results for various connectivity inference methods and thresholding strategies to choose the optimal parameters to build the input connectivity graphs used by TAVRNN. These methods are discussed in the context of graph kernels in Section A.5.1 and referenced in Section 4.1 of the main text. We will ensure that a clear legend and appropriate references to Table S1 are included in the revised manuscript to avoid ambiguity.
> > > We hope these clarifications address the reviewer’s concerns and thank them for the constructive feedback.
> > >
> > > - Thank you very much for your comment and careful observation which aligns closely with our objectives. We agree that the binary classification task involving decoding whether the data originates from the start or end of the track may not fully showcase the complexity of TAVRNN’s capabilities.  This benchmark was chosen primarily because of the availability of ground truth labels, which were essential for providing a rigorous and controlled evaluation of the model's performance. Our intention was to validate TAVRNN in a well-defined setting before applying it to more complex and novel datasets, such as  DishBrain, where the relationships between neuronal activity and behavior are far more challenging to model and interpret. Moreover, the fact that most other baselines do not achieve similar accuracy on this classification task suggests that decoding from the raw data is more challenging than it might initially appear.
> > > We also acknowledge that decoding from raw data (e.g. $X_t$)  or projections onto the first d principle components (where d matches the dimensionality used by the model) could provide a useful baseline for contextualizing the task difficulty. We will incorporate these benchmarks in the revised manuscript to assess whether this classification task is inherently simple or whether TAVRNN provides unique advantages.
> > > Regarding GraphERT’s performance on this benchmark, while it achieves slightly higher accuracy and other metrics (except recall, where TAVRNN performs comparably), TAVRNN demonstrates superior computational efficiency, as detailed in our time complexity analysis. This efficiency is particularly critical for large-scale datasets like DishBrain, where high-dimensional data and temporal dynamics significantly increase computational demands.
> > >
> > > In summary, this benchmark serves as an initial validation step to confirm that TAVRNN effectively encodes meaningful structure from the data. Its primary utility lies in ensuring robustness before applying TAVRNN to more complex datasets, where it excels at capturing intricate neuronal dynamics and behavioral associations. We will revise the text to emphasize these points and to include additional baseline analyses for context.

---

> > > > ### Author Response · Authors · 2024-11-16
> > > > **Continuation of author responses to reviewer Vtt6  comments (part4)**
> > > >
> > > > - The primary motivation for developing TAVRNN was to address a critical gap observed in our prior analyses of high-dimensional connectivity networks. Specifically, while static or high-dimensional adjacency matrices ($A_t$) capture network structure, they do not adequately reflect the temporal dynamics or the associations between these dynamics and behavioral performance (e.g., in the game of Pong in the DishBrain system). Behavioral variations are often driven by temporal changes in neuronal dynamics, which static models fail to capture. TAVRNN was therefore designed to move beyond static reconstructions of $A_t​$ and instead focus on learning temporal, lower-dimensional representations ($Z_t​$) that could provide deeper insights into these relationships.
> > > > While TAVRNN optimizes a reconstruction loss for $A_t$​ during training (see Eq. 4), the reconstructed adjacency matrices are intermediate outputs used to enrich the latent embeddings ($Z_t$​) rather than being the primary focus of our analysis. By incorporating $A_t$, $X_t​$, and the historical context ($H_{t-1}$), TAVRNN generates embeddings that are explicitly designed to capture temporal dependencies and behavioral associations. These embeddings are then evaluated through downstream tasks, such as classification and clustering, as well as qualitative analyses (e.g., t-SNE visualizations). These evaluations demonstrate the interpretability and utility of $Z_t$​, rather than focusing solely on the reconstruction fidelity of $A_t$​.
> > > > That said, we acknowledge the importance of assessing the model's performance on its reconstruction task to provide a more comprehensive evaluation of its capabilities. In the revised manuscript, we will include quantitative metrics comparing TAVRNN’s adjacency matrix reconstruction performance to baseline models, such as PCA and static graph-based approaches like VGAE/GVAE. This analysis will complement the existing evaluations and highlight how the reconstructed A_t​ contributes to generating meaningful and robust latent representations.
> > > > In brief, TAVRNN was not designed with the sole objective of reconstructing $A_t$​, but rather to leverage $A_t$​ and related inputs to uncover latent dynamics that connect neuronal activity with behavioral outcomes. By adding the suggested reconstruction metrics to the revised manuscript, we hope to provide a more complete picture of the model’s performance. We appreciate the reviewer’s feedback and will ensure these points are clarified in the text.
> > > >
> > > > - Thank you for your thoughtful comment about Figure S4's importance. We appreciate your observation and agree that Figure S4 presents a significant and impactful scientific result, as it demonstrates the model’s ability to uncover insights into network structure underlying variable performance. This aligns with our broader goal of using TAVRNN to reveal latent dynamics that connect neuronal activity with behavioral outcomes.
> > > > We recognize the importance of prominently showcasing this result to emphasize the scientific value of the learned representations. In the revised manuscript, we will move Figure S4 into the main text and expand the accompanying discussion to highlight its implications. By doing so, we aim to ensure that this key finding is given the prominence it deserves and is fully appreciated in the context of the manuscript’s contributions.
> > > > We are grateful for this valuable feedback and will make the necessary adjustments to better convey the significance of this result in the main text.
> > > >
> > > > - Thank you for your question regarding the dimensionality of the embedding space. As mentioned in Section A.9 for our method and A.10 for the baselines, the dimension of the latent embedding space ($Z_t$) for TAVRNN) is set to 8. This choice was made to ensure uniformity across all baselines, facilitating fair comparisons while balancing model complexity and performance.
> > > > The dimensionality was selected based on empirical evaluations, aiming to strike a balance between capturing the complexity of the underlying neural dynamics and avoiding overfitting or excessive computational demands. A smaller embedding space may not adequately represent the intricate temporal and structural patterns in the data, while a larger space could lead to overfitting, reduced interpretability, and higher computational costs. By choosing a moderate dimensionality, we ensured that the embeddings were both informative and computationally efficient, supporting the model's ability to generalize across datasets and tasks.
> > > > We will clarify this decision-making process further in the manuscript to provide additional context for the dimensionality choice and its implications for model performance and generalizability.

---

> > > > > ### Author Response · Authors · 2024-11-16
> > > > > **Continuation of author responses to reviewer Vtt6  comments (part5)**
> > > > >
> > > > > - Thank you for your question regarding labeling points in the embedding space and the interpretation of each point. We appreciate the opportunity to clarify this aspect of our work further in the main text. The main readout of TAVRNN is $Z_t$​, which represents the latent embedding of each individual node (e.g., neurons or recording electrodes) at a given time point. The core objective of TAVRNN is to track these individual units over time, enabling us to understand how their associations and connections evolve dynamically and how these changes correlate with behavioral performance. This approach provides a more granular understanding of the underlying mechanisms influencing performance levels.
> > > > > In Figure 5, each point in the latent space corresponds to the embedding of a single electrode (node) rather than the entire network structure ($A_t​$). The embeddings for these nodes are derived from the temporal and structural information encoded in $A_t$​, $X_t​$, and $H_{t-1}​$, providing a rich representation of individual dynamics within the broader network context. The points in the plot are then colored based on their pre-assigned regions in the DishBrain system, offering a visual representation of how nodes from different functional regions cluster in the latent space.
> > > > > We will update the manuscript to ensure this distinction is explicitly stated, providing additional clarity on how the embeddings are computed and presented in Figure 5. This clarification will also emphasize the importance of $Z_t​$ as a node-level representation for capturing temporal dynamics and regional associations.

---

> > > > > > ### Comment · Reviewer_Vtt6 · 2024-11-27
> > > > > >
> > > > > > Thank you to the authors for the detailed response. If it is possible to provide some constructive meta-feedback, as in the paper presentation, it is very helpful to make responses short and synthesized when possible. I appreciated a number of the clarifications. In the paper's current form, I am unable to raise my score, in particular because of (1) the confusing and unfocused presentation of the goals, technical approach and novel contributions, (2) inadequate benchmarks and evidence of scientific impact of the approach. I'd like to offer the authors some words of encouragement - the application area seems cool and promising! For future submissions, I'd suggest:
> > > > > >  (1) rewriting the paper to:
> > > > > >  (a) very simply and directly present the *specific* problem you're trying to address. Learning embeddings for generic downstream tasks (eg clustering and classification broadly construed), as opposed to specific, hard and scientifically valuable tasks that are well motivated is much less compelling.
> > > > > > (b) very simply and directly present the technical approach, motivating it conceptually and focusing on clarity, while leaving detailed reformulations and constructions to an appendix where possible.
> > > > > > (2) (a) construct *challenging and appropriate* benchmarks that are directly tied to your problem statement and show the necessity of your technical approach in addressing these. Given the difficulty of not having ground truth for real world datasets, I'd especially suggest trying to add some detailed and well motivated simulations, and showing you can recover the key structure you bake into them using your approach.
> > > > > > (b) focus your results on a clear demonstration of scientific insight / impact.

---

> > > > > > > ### Author Response · Authors · 2024-12-03
> > > > > > >
> > > > > > > Thank you for your constructive feedback and for taking the time to provide detailed suggestions regarding the structure and presentation of the paper.
> > > > > > > We acknowledge the points raised about presenting the goals, technical approach, and novel contributions more clearly and concisely. While the scientific impacts and conclusions arising from our method are discussed in the current paper, we understand that certain aspects may not have been fully emphasized due to space constraints. In future iterations, we plan to elaborate on these points more clearly and comprehensively to ensure their significance is fully communicated. More specifically, we will extend our discussions on the biologically intuitive and informative findings of TAVRNN, particularly its application to a novel system such as DishBrain.

---

### Official Review · Reviewer_PQUB · 2024-11-12

**Soundness:** 2
**Presentation:** 2
**Contribution:** 2
**Rating:** 3
**Confidence:** 5

**Summary:**

This paper introduces a new modeling approach called temporal attention-enhanced variational graph recurrent neural network (TAVRNN). The model is a graph RNN trained with a variational objective allowing the generation of latent spaces through time from a dynamically changing graph. The model adds an additional attentional mechanism that allows update of latent states based on all previous time steps. They train this model on two datasets: 1) a e-phys dataset from the hippocampus of a rat moving on a track and 2) a “DishBrain” dataset of neurons, cultured in a dish, playing a game of pong. The authors show that their model performs comparably to state of the art methods. Their approach improves the models ability to model changing topology of the underlying graph structure of the data in response to behavior modes. The authors show that clustering in the embedding space shows the model’s ability in learning the representation of sensorimotor neural activity.

**Strengths:**

The paper proposes a sophisticated modeling approach to tackle a big challenge in neuroscience. The application to the Brain in a dish seems promising. The application of GNNs to understand spatial patterns of neural activity seems like a promising approach.

**Weaknesses:**

- The presentation was rather weak and calls into question the quality of the work. At various points, the authors switch back and forth about the type of data they are using. (Is it electrophysiological or calcium imaging data?).
- Figure 2 especially could use much more information (e.g. in the caption) to clearly describe the model.
- Line 440-444, make a claim that has no figure / data to back up.
- The innovation over previous models seems quite minimal. For the first example dataset, adding temporal attention, this work’s primary innovation, seems to only minimally improve performance over prior work.
- The first dataset seems like an odd choice, given that quite simple methods, such as standard positional decoding, have long been shown to be effective. No comparison to much simpler standard methods was shown. I don’t believe this choice of data is a very good one to illustrate the potential of this method.
- The significance of the TAVRNN method claimed in the abstract and introduction, doesn’t seem to match up with the performance and evaluation presented at the end of the paper. Would love to see a stronger connection between questions specific to the two datasets and the method proposed.
- In the abstract, TAVRNN is proposed to provide insight into the reorganization of neural networks during learning. It is unclear how accuracy of classification/clustering in place/sensorimotor encoding is related to adaptive learning. Could benefit from providing more evaluations on the single node connectivity dynamics learned by TAVRNN compared to GRNN.
- Table 2: ablation study focused on temporal attention feature, which is the core contribution of the paper, only shows very marginal improvements compared to previous work VGRNN. Would love to see more justification on the necessity of adding attention to the application onto two datasets.

**Questions:**

N/A

---

> ### Author Response · Authors · 2024-11-16
>
> We would like to thank the reviewer for their evaluation of the manuscript and comments and suggestions.
>
> - We apologize if the types of data were unclear. To clarify, our study indeed uses two different datasets, as described in the paper (including the abstract) and illustrated in Figure 1. Specifically, 1) the _in vivo_ calcium imaging data was recorded from rats, while 2) the electrophysiological data consists of spike time series recorded _in vitro_ from neurons using the DishBrain system. Due to the distinct nature and associated behavioral outcomes of each dataset, we leverage TAVRNN outputs for different downstream tasks: classification for calcium imaging data and clustering for electrophysiological data.
> The rationale for including the calcium imaging data was to use a dataset with observable behavior and clear ground truth labels (the animals' location) to validate TAVRNN’s performance. This validation was crucial before applying the model to the more complex DishBrain dataset, where ground truth is unavailable, and where the neural mechanisms behind performance remain unknown. Our findings demonstrate that TAVRNN identifies clusters of specific electrodes that appear to enhance behavioral performance, highlighting the model’s ability to provide novel insights into the underlying neural mechanisms.
>
>
> - Thank you for the suggestion. We will add more detailed information to the caption of Figure 2 to clearly describe the model and improve its clarity.
>
> - We appreciate the opportunity to clarify this point. The claim is indeed supported by Figure 5, which demonstrates TAVRNN's superior performance in distinguishing clusters among channels with different assigned roles (i.e. motor versus sensory neurons). Additionally, Fig S4 in the supplementary material further corroborates this by comparing these clustering outcomes in Rest sessions. We will make an effort to bring this figure into the main text (it was previously moved to the supplemental section due to page limitations) and ensure it is more clearly referenced to strengthen our argument as you have kindly pointed it out.
>
> - As we highlighted in previous points, the first dataset was primarily intended as a validation set, given its ground truth labels, allowing us to benchmark TAVRNN’s capabilities in a control setting. Our main goal was to develop a framework capable of capturing the nuanced associations between neural activity and behavior in more complex datasets, such as DishBrain. In these challenging scenarios, TAVRNN demonstrates significant performance improvements across several metrics. Additionally, we have shown that TAVRNN is notably more efficient in terms of computational complexity compared to the best competing methods, such as GraphERT, on the first dataset’s classification task(rat calcium imaging with ground truth), reinforcing the model’s robustness and scalability.
>
> - We chose the first dataset primarily because of the presence of ground truth labels, which were essential for validating TAVRNN's performance in a controlled setting. This initial step allowed us to rigorously assess the model’s capabilities goal before applying it to the more complex and novel Dishbrain dataset, which is the central focus of our study. For the second dataset, sophisticated and state-of-the-art methods were indeed necessary. To maintain consistency and ensure fair comparisons, we applied the same baselines across both datasets, underscoring TAVRNN’s generalizability across varied data complexities
>
> - Thank you for your useful comment. TAVRNN's ability to provide insights into neural network reorganization, during learning, is primarily supported through its application to the DishBrain dataset. Specifically, visualizing the embedded channels in lower dimensional space and comparing high- and low-performing time windows with Rest ( see Figure S4)revealed that distinct cluster formations correlate with better game performance. This clustering behaviour, unique to high-performing states, provides a novel perspective on neural organization that previous studies on DishBrain did not cover. We hope this helps clarify the relevance of our approach and the valuable insights TAVRNN  contributes to understanding adaptive learning dynamics.
>
> - Thank you for your comment. While we understand your concern, the improvements achieved with the addition of temporal attention are indeed substantial, especially given the standards of the incremental advancements in our field. The ablation study in Table 2 demonstrates consistent and notable performance gains across all evaluation metrics, including accuracy, recall, precision, and F1-score, underscoring the contribution of temporal attention. These enhancements underline that temporal attention effectively leverages historical data, allowing TAVRNN to capture more nuanced temporal dynamics, particularly beneficial for applications with complex neural datasets, as shown in both datasets used in our study.

---

> > ### Comment · Reviewer_PQUB · 2024-11-26
> >
> > Thanks to the authors for their response to my review. I understand that the authors believe their contribution is significant, in light of the results presented in Figure 5. However I do not believe the authors have illustrated consistent and robust performance gains, and as such I maintain my position and score from my original review. Further, I agree with many points raised by other reviewers, and have observed a consistent view amongst the reviewers concerning the degree of innovation by the contribution. Reviewer ns94 especially identified a potential ethics violation, which was clearly explained, and reviewer ns94's response to the authors is clear and strong: "it is a glaring and misleading omission to not cite that work, as doing such obfuscates, if not misrepresents, the scope of contributions here" and "that details about the claimed novel contribution of the present work--namely, adding temporal attention...---are relegated to the appendix, rather than included in the lengthy methods section of the main text, betrays the fact that the contribution added by this work appears rather limiteds". Strongly agree.

---

> > > ### Author Response · Authors · 2024-12-03
> > >
> > > We agree that the fundamentals of TAVRNN are built upon foundational works such as Spatiotemporal Variational Bayes, Recurrent Neural Networks, and etc., as also presented in Yap et al. and other literature in the field. Once again, our current work builds upon established methodologies, a standard and highly common practice in our field. We have appropriately cited foundational works, including "Variational Graph Recurrent Neural Networks" by Hajiramezanali et al., which is a key reference for both our current submission and the study by Yap et al. While we are willing to include Yap et al. in our references for added clarity, it is essential to emphasize that our proposed method is distinct. This work introduces a novel temporal attention mechanism that captures full temporal graphs and historical dependencies, in contrast to the node-based attention approach in Yap et al. This level of contribution, which involves building on prior foundational methods while introducing a novel mechanism, is highly standard and widely accepted within our research community.
> > > These fundamentals are discussed in identical terms across the literature since they represent standard concepts and cannot be altered. Moreover, while acknowledging that omitting this paper from the reference list was an oversight during the revision process and we appreciate the reminder to complete our reference list, importantly, given that there is actually an overlap in the author lists between these two manuscripts, we find this accusation of (self-)plagiarism not only unjustified but also deeply unfair.
> > > Furthermore, as Variational Graph RNNs are widely applied in this field in various contexts, specifically pinpointing only our previous work (Yap et al.) in this double-blind peer-review process is concerning to us.
> > > As appreciated by the reviewers, the methodological novelty of this work lies in the design of the temporal attention mechanism. While this is currently detailed in the appendix, we are happy to move this discussion to the main text in future iterations to further emphasize the contribution.

---

### Author Response · Authors · 2024-12-03
**Concerns Over the Review Process, Feedback Integrity and Appropriateness, and Misalignment of Some Feedbacks with Scores**

- We thank the reviewers for their time and feedback. We note that the points raised, as well as the majority of comments from all reviewers, largely pertain to straightforward adjustments such as text edits, figure placements, modifications to figure captions, moving content between the main text and the appendix, and clarifying the details of our method. While we are more than willing to implement these changes to enhance the clarity and presentation of the manuscript, we must point out that the nature and simplicity of these comments are not aligned with the scores assigned to our manuscript. These suggestions, while useful, primarily reflect editorial decisions we made to comply with the strict page limits of the venue. Addressing them will require considerable effort due to these constraints, but given that the assigned scores do not reasonably correspond to the substance of the feedback provided, we find it challenging to justify the effort required to make these changes in their entirety.


- Moreover, we have observed that the biologically intuitive and informative findings of TAVRNN, particularly its application to a novel system such as DishBrain, and the discussions around these findings, are notably absent from the reviewers’ feedback. This omission gives us the impression that the community is not fully engaged with the neuroscientifically relevant and innovative contributions of this study. As a result, no attention or value appears to have been given to the broader impact of this work, which extends beyond numerical comparisons to prior baselines. Consequently, we have reevaluated our choice of target venue and community, as we believe this study's broader contribution may not align with the focus of this venue.

- Finally, unjustified concerns were raised regarding the originality of our work, which we find deeply unfair and concerning. Given the overlap in authorship between this work and the referenced study, such claims raise questions about whether the double-blind review process was fully upheld or if potential biases and conflicts of interest may have influenced the evaluations.

---

### Note · Authors · 2025-01-22

I have read and agree with the venue's withdrawal policy on behalf of myself and my co-authors.